# Chemotactic smoothing of collective migration

**Tapomoy Bhattacharjee[1†‡], Daniel B Amchin[2†], Ricard Alert[3,4†§], Jenna Anne Ott[2], Sujit Sankar Datta[2]\***

[1]The Andlinger Center for Energy and the Environment, Princeton University, Princeton, United States; [2]Department of Chemical and Biological Engineering, Princeton University, Princeton, United States; [3]Lewis-Sigler Institute for Integrative Genomics, Princeton University, Princeton, United States; [4]Princeton Center for Theoretical Science, Princeton University, Princeton, United States

**Abstract** Collective migration—the directed, coordinated motion of many self-propelled agents—is a fascinating emergent behavior exhibited by active matter with functional implications for biological systems. However, how migration can persist when a population is confronted with perturbations is poorly understood. Here, we address this gap in knowledge through studies of bacteria that migrate via directed motion, or chemotaxis, in response to a self-generated nutrient gradient. We find that bacterial populations autonomously smooth out large-scale perturbations in their overall morphology, enabling the cells to continue to migrate together. This smoothing process arises from spatial variations in the ability of cells to sense and respond to the local nutrient gradient—revealing a population-scale consequence of the manner in which individual cells transduce external signals. Altogether, our work provides insights to predict, and potentially control, the collective migration and morphology of cellular populations and diverse other forms of active matter.

**\*For correspondence:**
ssdatta@princeton.edu

[†]These authors contributed equally to this work

**Present address:** [‡]National Centre of Biological Sciences, Tata Institute of Fundamental Research, Bangalore, India; [§]Max Planck Institute for the Physics of Complex Systems, Dresden, Germany

## Editor's evaluation

The work provides new insights into the dual role of chemotactic sensing in both generating and controlling bacterial wave front patterns. Novel and elegant experimental techniques supported by computations using phenomenological models validate the hypothesis that chemotactic sensing smooths morphological variations; however, experiments suggest a richer picture than that predicted by the theory.

## Introduction

The flocking of birds, schooling of fish, herding of animals, and procession of human crowds are all familiar examples of collective migration. This phenomenon also manifests at smaller scales, such as in populations of cells and dispersions of synthetic self-propelled particles. In addition to being a fascinating example of emergent behavior, collective migration can be critically important—enabling populations to follow cues that would be undetectable to isolated individuals (*Camley, 2018*), escape from harmful conditions and colonize new terrain (*Cremer et al., 2019*), and coexist (*Gude et al., 2020*). Thus, diverse studies have sought to understand the mechanisms by which collective migration can arise.

Less well understood, however, is how collective migration persists after a population is confronted with perturbations. These can be external, stemming from heterogeneities in the environment (*Sándor et al., 2017*; *Morin et al., 2016*; *Wong et al., 2014*; *Chepizhko and Peruani, 2013*; *Chepizhko et al.,*

**eLife digest** Flocks of birds, schools of fish and herds of animals are all good examples of collective migration, where individuals co-ordinate their behavior to improve survival. This process also happens on a cellular level; for example, when bacteria consume a nutrient in their surroundings, they will collectively move to an area with a higher concentration of food via a process known as chemotaxis.

Several studies have examined how disturbing collective migration can cause populations to fall apart. However, little is known about how groups withstand these interferences. To investigate, Bhattacharjee, Amchin, Alert et al. studied bacteria called *Escherichia coli* as they moved through a gel towards nutrients.

The *E. coli* were injected into the gel using a three-dimensional printer, which deposited the bacteria into a wiggly shape that forces the cells apart, making it harder for them to move as a collective group. However, as the bacteria migrated through the gel, they smoothed out the line and gradually made it straighter so they could continue to travel together over longer distances.

Computer simulations revealed that this smoothing process is achieved by differences in how the cells respond to local nutrient levels based on their position. Bacteria towards the front of the group are exposed to more nutrients, causing them to become oversaturated and respond less effectively to the nutrient gradient. As a result, they move more slowly, allowing the cells behind them to eventually catch-up.

These findings reveal a general mechanism in which limitations in how individuals sense and respond to an external signal (in this case local nutrient concentrations) allows them to continue migrating together. This mechanism may apply to other systems that migrate via chemotaxis, as well as groups whose movement is directed by different external factors, such as temperature and light intensity.

---

*2013*; *Chepizhko and Peruani, 2015*; *Toner et al., 2018*; *Maitra, 2020*), or internal, stemming from differences in the behavior of individuals (*Yllanes et al., 2017*; *Bera and Sood, 2020*; *Alirezaeizanjani et al., 2020*). Mechanisms by which such perturbations can *disrupt* collective migration are well documented. Indeed, in some cases, perturbations can abolish coordinated motion throughout the population entirely (*Sándor et al., 2017*; *Morin et al., 2016*; *Yllanes et al., 2017*; *Bera and Sood, 2020*; *Chepizhko and Peruani, 2013*; *Chepizhko et al., 2013*; *Chepizhko and Peruani, 2015*; *Toner et al., 2018*). In other cases, perturbations couple to the active motion of the population to destabilize its leading edge, producing large-scale disruptions to its morphology (*Wong et al., 2014*; *Alert and Trepat, 2020*; *Alert et al., 2019*; *Driscoll et al., 2016*; *Doostmohammadi et al., 2016*; *Williamson and Salbreux, 2018*; *Miles et al., 2019*). Indeed, for one of the simplest cases of collective migration—via chemotaxis, the biased motion of cells up a chemical gradient—morphological instabilities can occur due to the disruptive influence of hydrodynamic (*Subramanian et al., 2011*; *Lushi et al., 2012*; *Lushi et al., 2018*) or chemical-mediated (*Ben Amar and Bianca, 2016*; *Ben Amar, 2016*; *Funaki et al., 2006*; *Brenner et al., 1998*; *Mimura and Tsujikawa, 1996*; *Stark, 2018*) interactions between cells. By contrast, mechanisms by which migrating populations can *withstand* perturbations have scarcely been examined.

Here, we demonstrate a mechanism by which collectively migrating populations of *Escherichia coli* autonomously smooth out large-scale perturbations in their overall morphology. We show that chemotaxis in response to a self-generated nutrient gradient provides both the driving force for collective migration and the primary smoothing mechanism for these bacterial populations. Using experiments on 3D-printed populations with defined morphologies, we characterize the dependence of this active smoothing on the wavelength of the perturbation and on the ability of cells to migrate. Furthermore, using continuum simulations, we show that the limited ability of cells to sense and respond to a nutrient gradient causes them to migrate at different velocities at different positions along a front—ultimately driving smoothing of the overall population and enabling continued collective migration. Our work thus reveals how cellular signal transduction enables a population to withstand large-scale perturbations and provides a framework to predict and control chemotactic smoothing for active matter in general.

## Results

### Chemotactic smoothing is regulated by perturbation wavelength and cellular motility

To experimentally investigate the collective migration of *E. coli* populations, we confine them within porous media of tunable properties (*Bhattacharjee and Datta, 2019a*; *Bhattacharjee and Datta, 2019b*; *Bhattacharjee et al., 2021*), as schematized in *Figure 1A and B* and detailed in Materials and methods. The media are composed of hydrogel particles that are swollen in a defined rich liquid medium with *L*-serine as the primary nutrient and chemoattractant. We enclose the particles at prescribed jammed packing fractions in transparent chambers. Because the hydrogel is highly swollen, it is freely permeable to oxygen and nutrient. However, while the particles do not hinder exposure of bacteria to these chemical signals, the cells cannot penetrate the individual particles and are instead forced to swim through the interparticle pores (*Figure 1B*). Varying the hydrogel particle packing density thus enables us to tune pore size and thereby modulate cellular migration without altering the nutrient field (*Bhattacharjee and Datta, 2019a*; *Bhattacharjee and Datta, 2019b*; *Bhattacharjee et al., 2021*). Specifically, we vary the mean pore size $\xi$ between 1.2 µm and 2.2 µm, causing cellular migration through the pore space to be more and less hindered, respectively, without deforming the solid matrix (*Bhattacharjee and Datta, 2019a*). Moreover, the packings are transparent, enabling the morphologies of the migrating populations to be tracked in the $xy$ plane using confocal fluorescence microscopy (*Figure 1A*); to this end, we use cells that constitutively express green fluorescent protein throughout their cytoplasm.

A key feature of the hydrogel packings is that they are yield-stress solids; thus, an injection micronozzle can move along a prescribed path inside each medium by locally rearranging the particles, gently extruding densely packed cells into the interstitial space (*Figure 1A and B*). The particles then rapidly re-densify around the newly introduced cells, re-forming a jammed solid matrix that supports the cells in place with minimal alteration to the overall pore structure (*Bhattacharjee et al., 2015*; *Bhattacharjee et al., 2016*; *Bhattacharjee et al., 2018*). This approach is therefore a form of 3D printing that enables the initial morphology of each bacterial population to be defined within the porous medium. The cells subsequently swim through the pores between particles, migrating outward through the pore space. For example, as we showed previously (*Bhattacharjee et al., 2021*), cells of *E. coli* initially 3D-printed in densely packed straight cylinders collectively migrate radially outward in smooth ('flat'), coherent fronts. These fronts form and propagate via chemotaxis: the cells continually consume surrounding nutrient, generating a local gradient that they in turn bias their motion along (*Adler, 1966*; *Cremer et al., 2019*; *Fu et al., 2018*; *Saragosti et al., 2011*; *Bai et al., 2021*). As each front of cells migrates, it propagates the local nutrient gradient with it through continued consumption, thereby sustaining collective migration. In the absence of nutrient, migrating fronts do not form at all (*Bhattacharjee et al., 2021*).

To test how perturbations in the overall morphology of the population influence its subsequent migration, we 3D-print densely packed *E. coli* in 1-cm-long cylinders with spatially periodic undulations as perturbations prescribed along the $x$ direction (*Figure 1B*). Each population is embedded deep within a defined porous medium; an initial population morphology is schematized at time $t = 0$ in *Figure 1B*, with the undulation wavelength and amplitude denoted by $\lambda$ and $A$, respectively. An experimental realization with $A(t = 0) \approx 300$ µm, $\lambda \approx 0.8$ mm, and $\xi = 1.7$ µm is shown in white in *Figure 1C*, which shows an $xy$ cross section through the midplane of the population. After 3D printing, the outer periphery of the population spreads slowly, hindered by cell-cell collisions in the pore space, as the population establishes a steep gradient of nutrient through consumption (*Bhattacharjee et al., 2021*). Then, this periphery spontaneously organizes into an ~300 µm-wide front of cells that collectively migrates outward (yellow in *Figure 1C*). The undulated morphology of this front initially retains that of the initial population. Strikingly, however, the front autonomously smooths out these large-scale undulations as it continues to propagate (*Video 1*). We characterize this behavior by tracking the decay of the undulation amplitude, normalized by its initial value $A_0 \equiv A(\Delta t = 0)$, as a function of time elapsed from the initiation of smoothing, $\Delta t$ (green circles in *Figure 1F*). The normalized amplitude decays exponentially (red line in *Figure 1F*), with a characteristic time scale $\tau \approx 2.5$ hr, and the population eventually continues to migrate as a completely flat front (cyan in *Figure 1C*).

We observe similar behavior when the wavelength $\lambda$ is increased to 3.4 mm (*Figure 1D*, *Video 2*) or when the pore size $\xi$ is increased to 2.2 µm (*Figure 1E*, *Video 3*); however, the dynamics of

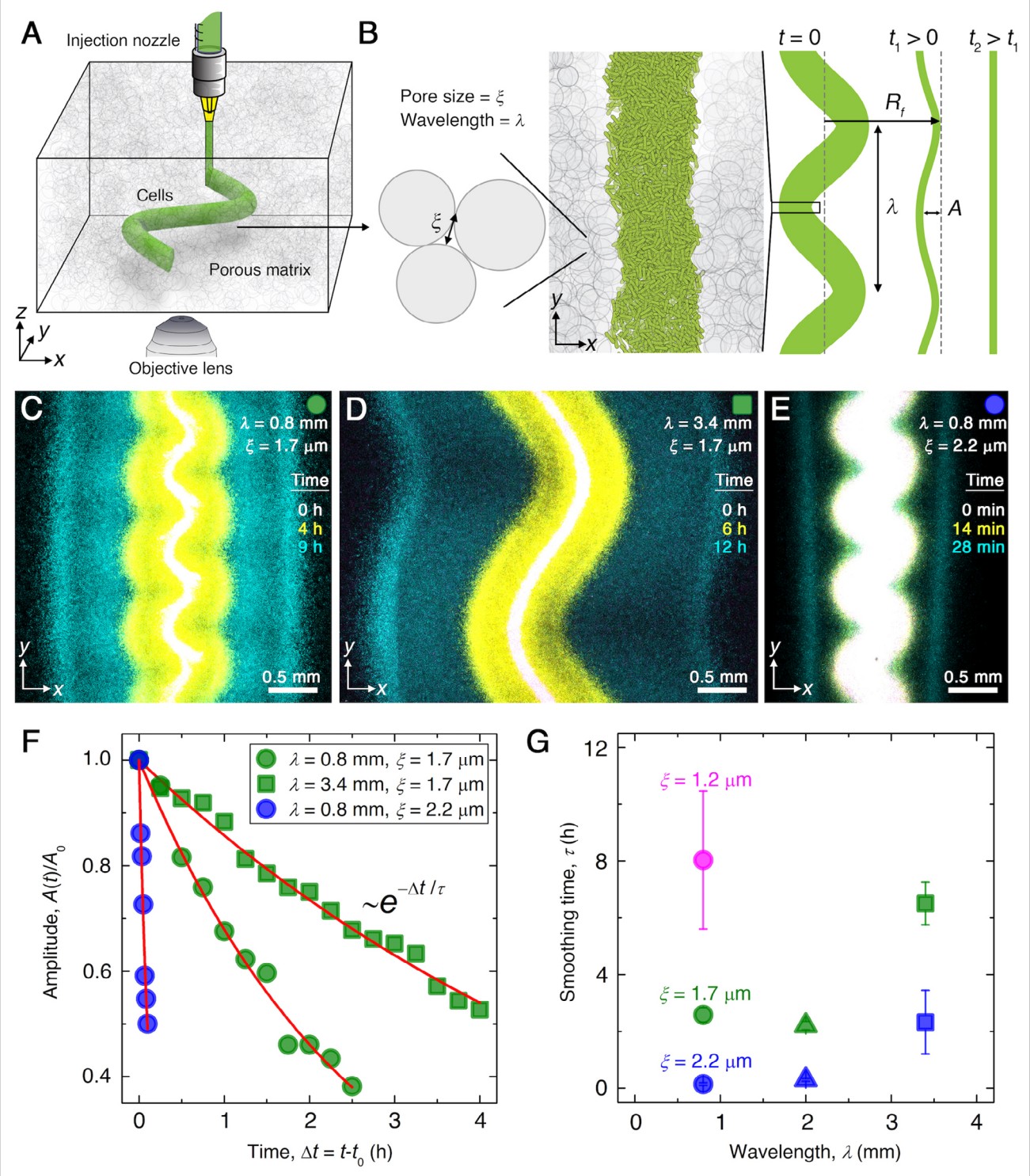

**Figure 1.** Experiments reveal that migrating *E. coli* populations autonomously smooth large-scale morphological perturbations. (**A**) Schematic of an undulated population (green cylinder) 3D-printed within a porous medium made of jammed hydrogel particles (gray). Each undulated cylinder requires ~10 s to print, two orders of magnitude shorter than the duration between successive 3D confocal image stacks, ~10 min. The surrounding medium fluidizes as cells are injected into the pore space, and then rapidly re-jams around the dense-packed cells. (**B**) Two-dimensional $xy$ slice through the midplane of the population. The starting morphology of the 3D-printed population has undulation wavelength $\lambda$ and amplitude $A_0$, as defined by the undulated path traced out by the injection nozzle. The cells subsequently swim through the pores between hydrogel particles, with mean pore size $\xi$. The population thereby migrates outward in a coherent front that eventually smooths; we track the radial position of the leading edge of the front $R_f$ and the undulation amplitude $A$ over time $t$. (**C–E**) Bottom-up ($xy$ plane) projections of cellular fluorescence intensity measured using 3D confocal

*Figure 1 continued on next page*

*Figure 1 continued*

image stacks. Images show sections of three initially undulated populations in three different porous media, each at three different times (superimposed white, yellow, cyan), as the cells migrate radially outward. A pixel corresponds to approximately one cell, and the images only show a magnified view of the overall population. Panels (**C**) and (**D**) demonstrate the influence of varying the undulation wavelength, keeping the mean pore size the same; increasing $\lambda$ slows smoothing. Panels (**C**) and (**E**) demonstrate the influence of varying the pore size, keeping the undulation wavelength the same; increasing $\xi$ hastens smoothing. (**F**) For each experiment shown in (**C–E**), the undulation amplitude $A$, normalized by its initial value $A_0$, decays exponentially with the time $\Delta t$ elapsed from the initiation of smoothing at $t = t_0$. Fitting the data (symbols) with an exponential decay (red lines) yields the smoothing time $\tau$ for each experiment. (**G**) Smoothing time $\tau$ measured in experiments increases with increasing undulation wavelength $\lambda$ and decreasing medium mean pore size $\xi$, which enables cells to migrate more easily. Error bars reflect the uncertainty in determining the initiation time $t_0$ from the exponential fit of the data.

front smoothing are altered in both cases. Specifically, increasing the undulation wavelength slows smoothing, increasing $\tau$ by a factor of $\approx 3$ to reach $\tau \approx 6.5$ hr (green squares in *Figure 1F*). Conversely, increasing the pore size—which enables cells to migrate through the pore space more easily—greatly hastens smoothing, decreasing $\tau$ by more than a factor of $\approx 10$ to become $\tau \approx 0.2$ hr (blue circles in *Figure 1F*). This behavior is consistent across multiple experiments with varying $\lambda$ and $\xi$, as summarized in *Figure 1G*. Our experiments thus indicate that the smoothing of collective migration is regulated by both the undulation wavelength and the ease with which cells migrate.

## A continuum model of chemotactic migration recapitulates the spatiotemporal features of smoothing

To gain further insight into the processes underlying smoothing, we use the classic Keller–Segel model of chemotactic migration (*Lauffenburger, 1991*; *Keller and Segel, 1971*) to investigate the dynamics of undulated populations. Variants of this model can successfully capture the key features of chemotactic migration of flat *E. coli* fronts in bulk liquid (*Keller and Segel, 1971*; *Fu et al., 2018*) and in porous media (*Bhattacharjee et al., 2021*); we therefore hypothesize that it can also help identify the essential physics of smoothing.

To this end, we consider a 2D representation of the population in the $xy$ plane for simplicity, with $\vec{r} \equiv (x, y)$, and model the evolution of the nutrient concentration $c(\vec{r}, t)$ and number density of bacteria $b(\vec{r}, t)$ using the coupled equations:

$$\partial_t c = D_\mathrm{c} \nabla^2 c - b \kappa g(c), \tag{1}$$

$$\partial_t b = -\nabla \cdot \vec{J}_\mathrm{b} + b \gamma g(c), \qquad \vec{J}_\mathrm{b} = -D_\mathrm{b} \nabla b + b \chi \nabla f(c). \tag{2}$$

*Equation 1* relates changes in $c$ to nutrient diffusion and consumption by the bacteria; $D_\mathrm{c}$ is the nutrient diffusion coefficient, $\kappa$ is the maximal consumption rate per cell, and $g(c) = c/\left(c + c_{1/2}\right)$ describes the influence of nutrient availability relative to the characteristic concentration $c_{1/2}$ through Michaelis–Menten kinetics. *Equation 2* relates changes in $b$ to the bacterial flux $\vec{J}_\mathrm{b}$, which arises from their undirected and directed motion, and net

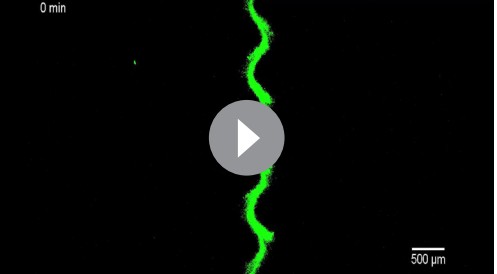

**Video 1.** Experiment probing chemotactic smoothing for $\lambda = 0.8$ mm, $\xi = 1.7$ μm. Video shows the maximum intensity fluorescence projection (bottom-up view) of migration from a 3D-printed undulated cylinder of closely packed *E. coli*. The cells collectively migrate outward in a front that autonomously smooths out the large-scale undulations as it continues to propagate.
https://elifesciences.org/articles/71226/figures#video1

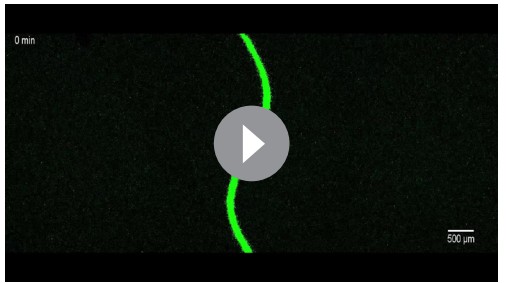

**Video 2.** Experiment probing chemotactic smoothing for $\lambda = 3.4$ mm, $\xi = 1.7$ μm. Video shows the maximum intensity fluorescence projection (bottom-up view) of migration from a 3D-printed undulated cylinder of closely packed *E. coli*.
https://elifesciences.org/articles/71226/figures#video2

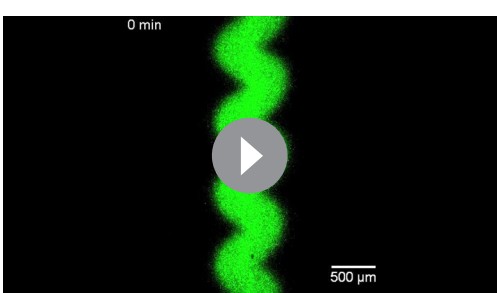

**Video 3.** Experiment probing chemotactic smoothing for $\lambda = 0.8$ mm, $\xi = 2.2$ µm. Video shows the maximum intensity fluorescence projection (bottom-up view) of migration from a 3D-printed undulated cylinder of closely packed *E. coli*.

https://elifesciences.org/articles/71226/figures#video3

cell proliferation with a maximal rate $\gamma$. In the absence of a nutrient gradient, bacteria move in an unbiased random walk (*Berg, 2004*); thus, undirected motion is diffusive over large length and time scales, with an effective diffusion coefficient $D_b$ whose value depends on both cellular activity and confinement in the pore space, and is therefore $b$-dependent as detailed in Materials and methods (*Bhattacharjee and Datta, 2019a*; *Bhattacharjee and Datta, 2019b*). In the presence of the local nutrient gradient established through consumption, bacteria perform chemotaxis, biasing this random walk (*Berg, 2004*); the function $f(c) \equiv \log\left[\left(1 + c/c_-\right)/\left(1 + c/c_+\right)\right]$ describes the ability of the bacteria to logarithmically sense nutrient with characteristic concentrations $c_-$ and $c_+$ (*Cremer et al., 2019*; *Fu et al., 2018*), and the chemotactic coefficient $\chi$ describes their ability to then bias their motion in response to the sensed nutrient gradient (*Keller and Segel, 1971*; *Fu et al., 2018*; *Cremer et al., 2019*). The chemotactic velocity is thus given by $\vec{v}_{ch} \equiv \chi \nabla f(c)$, where similar to $D_b$, the value of $\chi$ depends on both intrinsic cellular properties and pore-scale confinement, and is also $b$-dependent as detailed in Materials and methods (*Bhattacharjee et al., 2021*). Together, Equations 1 and 2 provide a continuum model of chemotactic migration that has thus far been successfully used to describe unperturbed *E. coli* populations (*Keller and Segel, 1971*; *Fu et al., 2018*; *Cremer et al., 2019*; *Bhattacharjee et al., 2021*). We note that a recently introduced *growth-expansion model* of chemotactic migration, for which analytical expressions describing chemotactic fronts have been obtained (*Cremer et al., 2019*; *Narla et al., 2021*), can be thought of as a limit of our model with bacterial growth taken to be independent of the attractant. An interesting direction for future work would be to study the phenomenon of chemotactic smoothing revealed here in the growth-expansion model, similar to a recent analytical study of small-amplitude perturbations in a simplified version of the model considered here (*Alert and Datta, 2021*).

Here, to simulate the chemotactic migration of populations with large-amplitude perturbations, we numerically solve Equations 1 and 2 using undulated morphologies as initial conditions for $b$, similar to those explored in the experiments. The simulations employ values for all parameters based on direct measurements, as detailed in Materials and methods. Although we do not expect perfect quantitative agreement between the experiments and simulations due to their difference in dimensionality and the simplified treatment of cell-cell interactions, the simulated fronts form, collectively migrate, and smooth in a manner that is remarkably similar to the experiments. Three examples are shown in *Figure 2C–E* (*Videos 4–6*), corresponding to the experiments shown in *Figure 1C–E* (*Videos 1–3*). Similar to the experiments, the outer periphery of each population first spreads slowly, then spontaneously organizes into an outward-migrating front that eventually smooths. We again find that the normalized undulation amplitude decays exponentially over time, as shown in *Figure 2D*. As in the experiments, increasing the undulation wavelength $\lambda$ slows smoothing; compare *Figure 2B* to *Figure 2A*. Also as in the experiments, increasing the pore size $\xi$, which increases the migration parameters $D_b$ and $\chi$, greatly hastens smoothing; compare *Figure 2C* to *Figure 2A*. This variation of the smoothing time scale $\tau$ obtained from simulations with $\lambda$ and $\xi$ is summarized in *Figure 2E*. We observe the same behavior as in the experiments, with the absolute values of $\tau$ agreeing to within a factor of ~3. This agreement confirms that the continuum Keller–Segel model recapitulates the essential spatio-temporal features of smoothing seen in the experiments.

## Chemotaxis is the primary driver of front smoothing

The simulations provide a way to directly assess the relative importance of cellular diffusion, chemotaxis, and cell proliferation to front smoothing. To this end, we perform the same simulation as in *Figure 2A*, but with each of the corresponding three terms in *Equation 2* knocked out, and determine

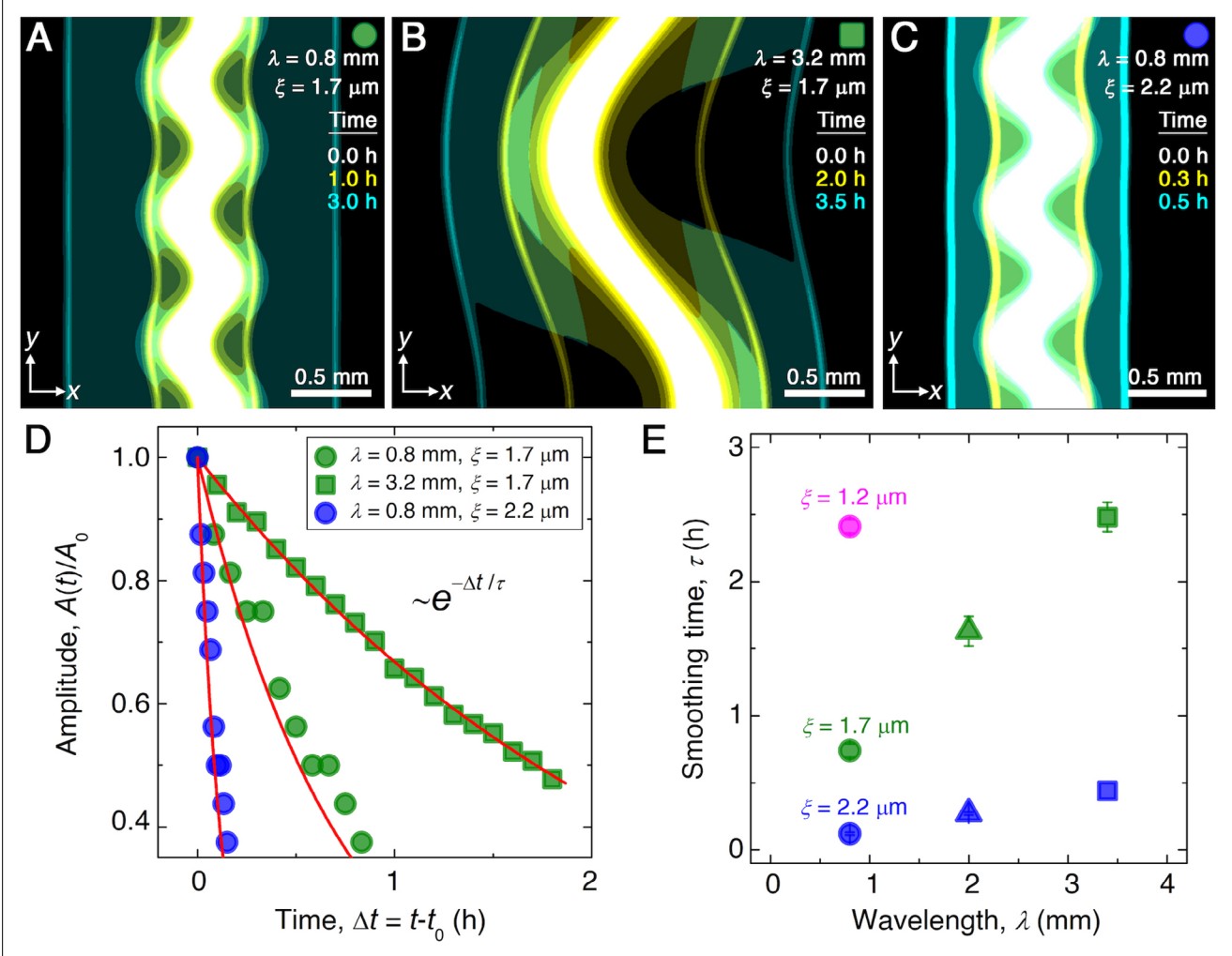

**Figure 2.** Continuum model captures the essential features of the smoothing of migrating bacterial populations. (**A–C**) Simulations corresponding to experiments reported in *Figure 1C and E*, respectively, performed by numerically solving Equations 1 and 2 in two dimensions ($xy$ plane). Images show the calculated cellular signal (details in Materials and methods) for three initially undulated populations in three different porous media, each at three different times (superimposed white, yellow, cyan), as the cells migrate outward. Panels (**A**) and (**B**) demonstrate the influence of varying the undulation wavelength, keeping the mean pore size the same; as in the experiments, increasing $\lambda$ slows smoothing. Panels (**A**) and (**C**) demonstrate the influence of varying the pore size, keeping the undulation wavelength the same; as in the experiments, increasing $\xi$, incorporated in the model by using larger values of the diffusion and chemotactic coefficients as obtained directly from experiments, hastens smoothing. (**D**) For each simulation shown in (**A–C**), the undulation amplitude $A$, normalized by its initial value $A_0$, decays exponentially with the time $\Delta t$ elapsed from the initiation of smoothing at $t = t_0$ as in the experiments. Fitting the data (symbols) with an exponential decay (red lines) again yields the smoothing time $\tau$ for each simulation. (**E**) Smoothing time $\tau$ obtained from the simulations increases with increasing undulation wavelength $\lambda$ and decreasing medium mean pore size $\xi$, as in the experiments. Error bars reflect the uncertainty in determining the initiation time $t_0$ from the exponential fit of the data.

the resulting impact on collective migration. This procedure enables us to determine the factors necessary for smoothing.

While diffusion typically causes spatial inhomogeneities to smooth out, we do not expect it to play an appreciable role in the front smoothing observed here: the characteristic time scale over which undulations of wavelength $\lambda \approx 1$ mm diffusively smooth is $\sim \lambda^2/D_b \approx 100$ to 700 hr, up to three orders of magnitude larger than the smoothing time $\tau$ measured in experiments and simulations. We therefore expect that the undirected motion of bacteria is much too slow to contribute to front smoothing. The simulations for $\lambda = 0.8$ mm and $\xi = 1.7$ µm confirm this expectation: setting $D_b = 0$ yields fronts that still smooth over a time scale $\tau \sim 1$ hr similar to the full simulations (*Figure 3A*).

Another possible mechanism of front smoothing is differences in bacterial proliferation at different locations along the front periphery—for example, the front would smooth if cells in concave regions

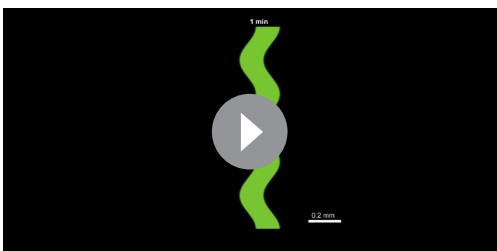

**Video 4.** Simulation probing chemotactic smoothing for $\lambda = 0.8$ mm, $\xi = 1.7$ μm. Video shows the calculated cellular fluorescence signal of cells migrating from an undulated stripe of closely packed *E. coli* similar to Video 1. As in the experiments, the cells collectively migrate outward in a front that autonomously smooths out the large-scale undulations as it continues to propagate.

https://elifesciences.org/articles/71226/figures#video4

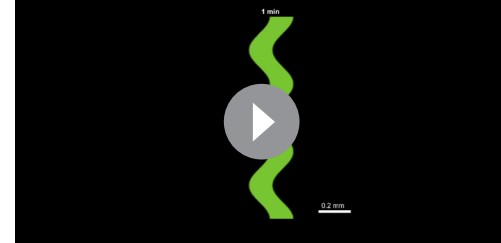

**Video 6.** Simulation probing chemotactic smoothing for $\lambda = 0.8$ mm, $\xi = 2.2$ μm. Video shows the calculated cellular fluorescence signal of cells migrating from an undulated stripe of closely packed *E. coli* similar to Video 3.

https://elifesciences.org/articles/71226/figures#video6

were able to proliferate faster than those in convex regions. However, differential proliferation typically destabilizes bacterial communities, as shown previously both experimentally and theoretically (*Fujikawa and Matsushita, 1989*; *Bonachela et al., 2011*; *Nadell et al., 2010*; *Farrell et al., 2013*; *Trinschek et al., 2018*; *Allen and Waclaw, 2019*). Furthermore, even if proliferation were to help smooth the overall population, we again expect this hypothetical mechanism to be too slow to appreciably contribute: the shortest time scale over which cells all growing exponentially at a maximal rate $\gamma \sim 1$ hr$^{-1}$ spread over the length scale $A_0 \approx 300$ μm by growing end-to-end is $\gamma^{-1} \log_2 \left( A_0/l_{\text{cell}} \right) \sim 7$ hr, where $l_{\text{cell}} \approx 2$ μm is the cell body length. This time scale is over an order of magnitude larger than the $\tau$ measured in experiments and simulations. The simulations again confirm our expectation: setting $\gamma = 0$ yields fronts that still smooth over a time scale $\tau \sim 1$ hr similar to the full simulations (*Figure 3B*).

These findings leave chemotaxis as the remaining possible mechanism of front smoothing. The simulations confirm this expectation: setting $\chi = 0$ yields a population that slowly spreads via diffusion and proliferation, but that does not form collectively migrating fronts at all (*Figure 3C*). Therefore, chemotaxis is both necessary and sufficient for the observed front smoothing.

## Distinct modes by which chemotaxis impacts front morphology

How exactly does chemotaxis smooth bacterial fronts? To address this question, we examine the spatially varying chemotactic velocity $\vec{v}_c = \chi \nabla f(c)$, which quantifies how rapidly different regions of the population migrate via chemotaxis. To gain intuition for the determinants of $\vec{v}_c$, we recast this expression in terms of the nutrient gradient:

$$\vec{v}_c = \underbrace{\chi f'(c)}_{\text{Response function}} \underbrace{\nabla c}_{\text{Forcing}}.$$ (3)

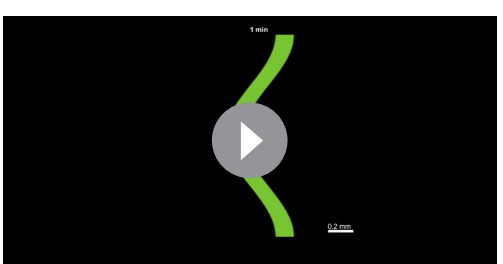

**Video 5.** Simulation probing chemotactic smoothing for $\lambda = 3.4$ mm, $\xi = 1.7$ μm. Video shows the calculated cellular fluorescence signal of cells migrating from an undulated stripe of closely packed *E. coli* similar to Video 2.

https://elifesciences.org/articles/71226/figures#video5

As in linear response theory, the chemotactic velocity can be viewed as the bacterial response to the driving force given by the nutrient gradient, $\nabla c$, modulated by the chemotactic response function $\chi f'(c)$. Thus, variations in chemotactic velocity along the leading edge of the front, which specify how the overall front morphology evolves, are determined by the combined effect of variations in the nutrient gradient and the chemotactic response function. We therefore examine each of these modes by which chemotaxis influences front morphology in turn.

We first consider the nutrient gradient, which is the typical focus of chemotaxis studies. Our

**Figure 3.** Chemotaxis is the primary driver of morphological smoothing. Images show the same simulation as in *Figure 2A*, which serves as an exemplary case, but with either (**A**) diffusive cell motion, (**B**) cell proliferation, or (**C**) cell chemotaxis knocked out by setting the diffusivity $D_b$, proliferation rate $\gamma$, or chemotactic coefficient $\chi$ to zero, respectively. Simulated bacterial fronts lacking diffusion or proliferation still smooth, as shown in (**A, B**), but simulated fronts lacking chemotaxis do not smooth, as shown in (**C**), demonstrating that chemotaxis is necessary and sufficient for the observed morphological smoothing.

simulations, which numerically solve the coupled system of Equations 1 and 2, directly yield the spatially varying nutrient field $c$ and therefore $\nabla c$. A snapshot from the representative example of *Figure 2A* is shown in *Figure 4A*, with the contours of $c = c_-$ and $c = c_+$ indicated by the cyan and magenta lines, respectively. The contours are spaced closer at the convex 'peaks' (e.g., at $y/\lambda = 0.5$) than at the concave 'valleys' (e.g., at $y/\lambda = 0$) along the leading edge of the front. Thus, the magnitude of the driving force given by $\nabla c$ is larger at the peaks. We confirm this expectation by directly quantifying the nutrient gradient along the leading edge, focusing on the component $\partial_x c$ in the overall front propagation direction ($x$) for simplicity, as shown by the orange symbols in *Figure 4C*; as expected, this driving force is stronger at the peaks. This spatial variation in the driving force promotes faster outward chemotactic migration at the peaks than at the valleys, amplifying front undulations—in opposition to our observation that the migrating population self-smooths. Variations in the local nutrient gradient along the leading edge of the front do not contribute to smoothing; rather, they oppose it.

We next turn to the chemotactic response function, which characterizes cellular signal transduction. Because $\chi$ is a constant for each porous medium (*Bhattacharjee et al., 2021*), spatial variations in the response function are set by variations in $f'(c)$. The sensing function $f(c)$ is plotted in the upper panel of *Figure 4B*. It varies linearly as $\sim c\,(1/c_- - 1/c_+)$ for $c \ll c_-$ and saturates at $\log\,(c_+/c_-)$ for $c \gg c_+$; the characteristic concentrations $c_-$ and $c_+$ represent the dissociation constants of the nutrient for the inactive and active conformations of the cell-surface receptors, respectively (*Cremer et al., 2019*; *Fu et al., 2018*; *Dufour et al., 2014*; *Yang et al., 2015*). The response function $\chi f'(c)$ therefore decreases strongly as $c$ increases above $c_+$, which accordingly is often referred to as an upper limit of sensing (*Figure 4B*, lower panel). That is, because high nutrient concentrations saturate cell-surface receptors, the chemotactic response function decreases with nutrient concentration. Inspection of the nutrient field indicates that nutrient concentrations are larger at the peaks than at the valleys along the leading edge of the front (*Figure 4A*). Thus, the chemotactic response of cells is weaker at peaks than at valleys, as shown by the points in *Figure 4B*, yielding slower outward chemotactic migration at peaks than at valleys and thereby reducing the amplitude of front undulations. Variations in the chemotactic response along the leading edge of the front promote smoothing, unlike variations in the nutrient gradient.

## Spatial variations in chemotactic response drive morphological smoothing

We therefore hypothesize that the stabilizing effect of the chemotactic response (*Figure 4C*, blue) dominates over the destabilizing influence of the nutrient gradient (*Figure 4C*, red), leading to smoothing. Computation of the spatially varying chemotactic velocity at the leading edge of the front using *Equation 3*, focusing on the $x$ velocity component $v_{c,x} \approx \chi f' \partial_x c$ for simplicity, supports this

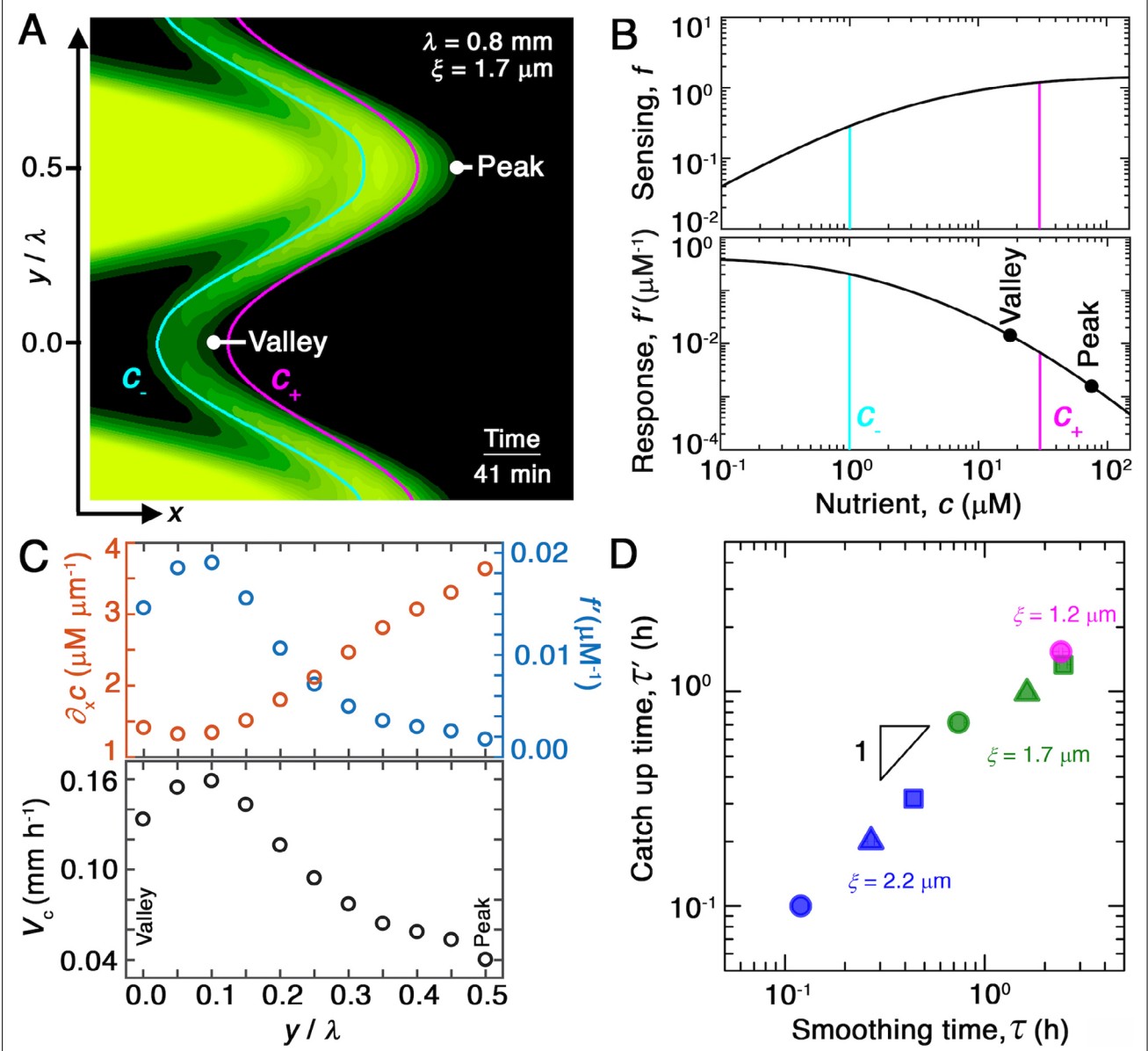

**Figure 4.** Chemotaxis alters the morphology of migrating bacterial fronts in two distinct ways. (**A**) Magnified view of a migrating bacterial front from the simulation shown in *Figure 2A* at time $t = 41$ min as a representative example. To illustrate the spatially varying nutrient levels, we show the contours of constant nutrient concentration $c = c_+$ and $c = c_-$ in magenta and cyan, respectively; these represent characteristic upper and lower limits of sensing. The contours are spaced closer at the leading edge of the convex peak ($y/\lambda = 0.5$) than the concave valley ($y/\lambda = 0$), indicating that the magnitude of the local nutrient gradient is larger at peaks than at valleys. The nutrient concentration itself, which increases monotonically with increasing $x$, is also larger at the peak than at the valley. (**B**) Top and bottom panels show the variation of the nutrient sensing function $f(c)$ and chemotactic response function $f'(c)$, respectively, with nutrient concentration $c$. Because sensing saturates at high nutrient concentrations, chemotactic response is weaker at higher $c$ (peaks) than at lower $c$ (valleys). (**C**) Top panel shows the $x$ component of the nutrient gradient $\partial_x c$ (red, left axis) and the response function $f'$ (blue, right axis), and bottom panel shows the $x$ component of the chemotactic velocity $v_{c,x} = \chi f' \partial_x c$ computed from these quantities, evaluated at different lateral positions $y$ along the leading edge of the front in (**A**). While the driving force of chemotaxis represented by $\partial_x c$ is smaller at the valley, the chemotactic response $\chi f'$ is larger at the valley and dominates in setting $v_{c,x}$: valleys move out faster than peaks, eventually catching up to them and smoothing out the undulations. (**D**) For all simulations (*Figure 2E*), the smoothing time $\tau$ determined by analyzing the decay of large-scale undulations (*Figure 2D*) is similar to the time $\tau'$ needed for valleys to catch up to peaks estimated using their different $x$-component chemotactic velocities. Note that we do not expect an exact match between $\tau$ and $\tau'$ as they are related yet different quantities.

The online version of this article includes the following figure supplement(s) for figure 4:

**Figure supplement 1.** Effect of reduced sensing.

**Figure supplement 2.** Chemotactic smoothing requires a concave sensing function $f(c)$.

*Figure 4 continued on next page*

*Figure 4 continued*

**Figure supplement 3.** Convergence of the numerical simulations.

hypothesis: cells at concave regions migrate outward faster than those at convex regions (*Figure 4C*, lower panel). To further test this hypothesis, we assess the influence of varying $c_+$; we expect that reducing this upper limit weakens chemotactic response not just at the peaks, but also the valleys, thereby slowing smoothing. While tuning solely $c_+$ is challenging in the experiments, this can be readily done in the simulation—yielding slower smoothing, as expected (*Figure 4—figure supplement 1*).

As a final test of our hypothesis, for each simulation shown in *Figure 2*, we determine the difference between the chemotactic velocities of the valleys and peaks, approximated by $\Delta v_{c,x} \approx \chi[(f'\partial_x c)_{valley} - (f'\partial_x c)_{peak}]$, as a function of time $\Delta t$. If smoothing is indeed due to variations of the chemotactic velocity along the leading edge, then the smoothing time $\tau$ determined by analyzing the decay of the undulation amplitude, $A = A_0 e^{-\Delta t/\tau}$ (*Figure 2D and E*), should be similar to the time $\tau'$ at which valleys catch up to peaks, that is, $\int_0^{\tau'} \Delta v_{c,x} \, \Delta t \approx A_0$. To test this expectation, we plot the $\tau'$ values thus obtained for all of our simulations of varying $\lambda$ and $\xi$ as a function of the corresponding $\tau$, as shown in *Figure 4D*. We find that $\tau'$ and $\tau$ are indeed similar to one another in all cases—confirming that smoothing is determined by spatial variations in chemotactic velocity.

## Discussion

By combining experiments and simulations, this study elucidates a mechanism by which collectively migrating populations can smooth out large-scale perturbations in their overall morphology. We focus on the canonical example of chemotactic migration, in which coherent fronts of cells move in response to a self-generated nutrient gradient. The smoothing of these fronts underlies the utility of standard agar-based assays for chemotaxis, in which bacteria spread outward in smooth, circular rings from a dense inoculum (*Wolfe and Berg, 1989*; *Tittsler and Sandholzer, 1936*; *Croze et al., 2011*; *Cremer et al., 2019*)—despite the presence of irregularities in the initial inoculum that are inevitably introduced by human error. To our knowledge, the robustness of the front morphology to such perturbations has never been examined or quantitatively explained; as a result, previous studies have only focused on the migration of the smooth fronts that ultimately result (*Adler, 1966*; *Lauffenburger, 1991*; *Keller and Segel, 1971*; *Cremer et al., 2019*; *Fu et al., 2018*; *Saragosti et al., 2011*; *Bhattacharjee et al., 2021*; *Bai et al., 2021*). Our work now provides an explanation for why perturbed fronts smooth out. It therefore provides a counterpoint to previous studies investigating the ability of perturbations to instead disrupt collective migration (*Sándor et al., 2017*; *Morin et al., 2016*; *Yllanes et al., 2017*; *Bera and Sood, 2020*; *Chepizhko and Peruani, 2013*; *Chepizhko et al., 2013*; *Chepizhko and Peruani, 2015*; *Toner et al., 2018*; *Wong et al., 2014*; *Alert and Trepat, 2020*; *Alert et al., 2019*; *Driscoll et al., 2016*; *Doostmohammadi et al., 2016*; *Williamson and Salbreux, 2018*; *Miles et al., 2019*; *Subramanian et al., 2011*; *Lushi et al., 2012*; *Lushi et al., 2018*; *Ben Amar and Bianca, 2016*; *Ben Amar, 2016*; *Funaki et al., 2006*; *Brenner et al., 1998*; *Mimura and Tsujikawa, 1996*; *Stark, 2018*). It also complements recent theoretical work describing how chemotaxis can stabilize the hydrodynamic instabilities that arise in unconfined populations of self-propelled particles (*Nejad and Najafi, 2019*).

The 3D printing platform provides a unique way to tune the shape of the initial perturbation, as well as the extent to which cellular migration is hindered. Our experiments using this approach reveal that the dynamics of smoothing are regulated by both the undulation wavelength and the ease with which cells migrate. The continuum simulations recapitulate the essential features of this behavior and shed light on the underlying mechanism. We find that even though cells in peaks of an undulated front experience a stronger driving force given by the local nutrient gradient, the higher nutrient levels they are exposed to saturate their cell-surface receptors, and hence they exhibit a weaker chemotactic response than cells in valleys. That is, while variations in the nutrient gradient along the leading edge of a front act to amplify undulations, variations in the ability of cells to sense and respond to this gradient dominate and instead smooth out the undulation. Importantly, this mechanism of smoothing is distinct from diffusion, which is typically responsible for the smoothing of traveling waves in reaction-diffusion systems—and in our case, is much too slow to drive smoothing.

## Conditions for chemotactic smoothing to arise

While our study utilizes a specific form of the sensing function $f(c)$ established for *E. coli* (*Cremer et al., 2019*; *Fu et al., 2018*), the phenomenon of chemotactic smoothing can manifest more generally. Specifically, our description of smoothing requires that (i) convex regions of a population are exposed to more nutrient $c$ than concave regions, and (ii) $f(c)$ is monotonically increasing and concave, with $f''(c) < 0$; when these conditions are satisfied, the chemotactic response is weaker at convex regions than at concave ones, thereby promoting smoothing (as indicated in *Figure 4B*).

The first requirement is frequently satisfied for collective migration in general; for example, in chemotactic migration, nutrient concentration $c$ decreases from the outward boundaries into the population over a length scale given by the interplay between nutrient diffusion and consumption. This first requirement is also satisfied by many other forms of active matter that rely on other modes of sensing to collectively migrate, for which $c$ would generically represent the stimulus being sensed. Documented examples include durotactic cell groups (*Roca-Cusachs et al., 2013*; *Sunyer et al., 2016*; *Alert and Casademunt, 2019*), phoretic active colloids (*Illien et al., 2017*; *Liebchen and Löwen, 2018Stark, 2018*), and phototactic robots (*Mijalkov et al., 2016*; *Palagi and Fischer, 2018*)—systems for which migration is directed toward regions of larger $c$, and therefore convex regions are more likely to be exposed to larger $c$.

The second requirement is also satisfied for diverse active matter systems; in the context of chemotaxis, specific examples include other bacteria (*Menolascina et al., 2017*), enzymes (*Jee et al., 2018*; *Agudo-Canalejo et al., 2018*; *Mohajerani et al., 2018*), aggregating amoeba cells (*Keller and Segel, 1970*), and mammalian cell groups during development, immune response, and disease (*Camley, 2018*; *Iglesias and Devreotes, 2008*; *Theveneau et al., 2010*; *McLennan et al., 2012*; *Malet-Engra et al., 2015*; *Puliafito et al., 2015*; *Tweedy et al., 2020*). This second requirement is again also satisfied for active matter that collectively migrates using other sensing mechanisms, for which sensing has been documented to increase and eventually saturate with the stimulus, be it the stiffness of the underlying surface (*Roca-Cusachs et al., 2013*; *Sunyer et al., 2016*; *Alert and Casademunt, 2019*), temperature (*Illien et al., 2017*; *Liebchen and Löwen, 2018*), or light intensity (*Mijalkov et al., 2016*; *Palagi and Fischer, 2018*). Thus, exploring the physics described here in diverse other forms of active matter will be a useful direction for future work.

As a final illustration of the necessity of the sensing function $f(c)$ to be concave, $f''(c) < 0$, we repeat our analysis but instead consider a strictly linear $f(c) = c/c_{\text{lin}}$, which does not saturate. We choose $c_{\text{lin}} = \left(1/c_- - 1/c_+\right)^{-1}$ so that the linear $f(c)$ matches our original logarithmic $f(c)$ at small $c$. With this linear sensing function, the chemotactic response is independent of concentration, $f'(c) = 1/c_{\text{lin}}$, and the condition of concavity is violated: $f''(c) = 0$. We therefore expect chemotactic smoothing to not occur. Consistent with our expectation, repeating the analysis underlying *Figure 4C* but for the strictly linear $f(c)$ yields fronts for which valleys no longer move faster than peaks. Instead, as shown in *Figure 4—figure supplement 2*, the profile of chemotactic velocity is now inverted with respect to that of the bottom panel in *Figure 4C*. Hence, the front does not smooth. Overall, this sample computation illustrates a way of modifying $f(c)$ that abrogates sensing saturation and hence would prevent chemotactic smoothing.

## Broader implications of chemotactic smoothing

The chemotactic smoothing process described here is autonomous, arising without any external intervention. Instead, it is a population-scale consequence of the limitations in cellular signal transduction—motivating future studies of other population-scale effects, beyond smoothing, that may emerge from individual behaviors. Indeed, while studies of chemotaxis typically focus on the role of the external nutrient gradient in driving cellular migration, our work highlights the distinct and pivotal role played by the cellular chemotactic response function in regulating migration and large-scale population morphology more broadly. Our work therefore contributes a new factor to be considered in descriptions of morphogenesis, which thus far have focused on the role of other factors—such as differential forcing by signaling gradients, differential proliferation, intercellular mechanics, substrate interactions, and osmotic stresses (*McLennan et al., 2012*; *Fujikawa and Matsushita, 1989*; *Bonachela et al., 2011*; *Nadell et al., 2010*; *Farrell et al., 2013*; *Trinschek et al., 2018*; *Allen and Waclaw, 2019*; *Beroz et al., 2018*; *Fei et al., 2020*; *Yan et al., 2019*; *Yan et al., 2017*; *Copenhagen et al., 2020*;

*Smith et al., 2017*; *Ghosh et al., 2015*; *Zhang et al., 2021*)—in regulating the overall morphology of cellular communities and active matter in general.

## Materials and methods

### Preparing and characterizing porous media

We prepared 3D porous media by dispersing dry granules of crosslinked acrylic acid/alkyl acrylate copolymers (Carbomer 980, Ashland) in liquid EZ Rich, a defined rich medium for *E. coli*. The components to prepare the EZ Rich were purchased from Teknova Inc, autoclaved prior to use, and were mixed following the manufacturer's directions; specifically, the liquid medium was an aqueous solution of 10× MOPS Mixture (M2101), 10× ACGU solution (M2103), 5× Supplement EZ solution (M2104), 20% glucose solution (G0520), 0.132 M potassium phosphate dibasic solution (M2102), and ultrapure Milli-Q water at volume fractions of 10, 10, 20, 1, 1, and 58%, respectively. We ensured homogeneous dispersions of swollen hydrogel particles by mixing each dispersion for at least 2 hr at 1600 rpm using magnetic stirring and adjusted the pH to 7.4 by adding 10 N NaOH to ensure optimal cell viability. The hydrogel granules swell considerably, resulting in a jammed medium made of ~5–10 µm diameter swollen hydrogel particles with ~20% polydispersity and with an individual mesh size of ~40–100 nm, as we established previously (*Bhattacharjee and Datta, 2019b*), which enables small molecules (e.g., amino acids, glucose, oxygen) to freely diffuse throughout the medium.

Tuning the mass fraction of dispersed hydrogel particles enables the sizes of the pores between particles to be precisely tuned. We measured the smallest local pore dimension by tracking the diffusion of 200-nm-diameter fluorescent tracers through the pore space, as we detailed in a previous paper (*Bhattacharjee et al., 2021*). This previous paper shows the full pore size distributions thereby measured for porous media prepared in an identical manner to those used here; in this paper, we only describe each medium using the mean pore size $\xi$, for simplicity. Indeed, the measured pore size distributions exhibit exponential decays characterized by the mean value $\xi$, as reported in *Bhattacharjee et al., 2021*, with pore sizes between 1 and 8 µm in the loosest packings and pores smaller than 4 µm in the tightest packings.

### 3D printing bacterial populations

Prior to each experiment, we prepared an overnight culture of *E. coli* W3110 in LB media at 30°C. We then incubated a 1% solution of this culture in fresh LB media for 3 hr until the optical density reached ~0.6, and then resuspended the cells in liquid EZ Rich to a concentration of $8.6 \times 10^{10}$ cells/mL. We then used this suspension as the inoculum that was 3D printed into a porous medium using a pulled glass capillary with a ~100–200 µm-wide opening as an injection nozzle. Each porous medium had a large volume of 4 mL and was confined in a transparent-walled glass-bottom Petri dish 35 mm in diameter and 10 mm in height; in each experiment, the injection nozzle was mounted on a motorized translation stage that traces out a programmed two-dimensional undulating path within the porous medium, at least ~500–1000 m away from any boundaries, at a constant speed of 1 mm/s. As the injection nozzle moved through the medium, it locally rearranged the hydrogel packing and gently extruded the cell suspension into the interstitial space using a flow-controlled syringe pump at 50 µL/hr, which corresponds to a gentle shear rate of ~4–36 s⁻¹ at the tip of the injection nozzle. As the nozzle continued to move, the surrounding hydrogel particles rapidly densified around the newly introduced cells, re-forming a jammed solid matrix (*Bhattacharjee et al., 2018*; *Bhattacharjee et al., 2015*; *Bhattacharjee et al., 2016*) that compressed the cellular suspension until the cells are closely packed to an approximate density of $0.95 \times 10^{12}$ cells/mL. This protocol thus results in a 3D-printed bacterial population having a defined initial amplitude and wavelength. Moreover, as we showed in our previous work *Bhattacharjee et al., 2021*, this process does not appreciably alter the properties of the hydrogel packing and is sufficiently gentle to maintain the viability and motility of the cells.

### Imaging bacteria within porous media

Because the 3D-printed undulated cylinders of densely packed cells are ~1 cm long, each printing process requires ~10 s. After 3D printing, the top surface of the porous medium was sealed with a thin layer of 1–2 mL of paraffin oil to minimize evaporation while allowing unimpeded oxygen diffusion. We then commenced imaging within a few minutes after printing. Once an undulated population is

3D printed, it maintains its shape until cells start to move outward through the pore space. The time needed to print each cylinder is two orders of magnitude shorter than the duration between successive 3D confocal image stacks. Moreover, the 3D printing is fast enough to be considered as instantaneous when compared with the speed of bacterial migration. Thus, the imaging is sufficiently fast to capture the front propagation dynamics. To image how the distribution of cells evolves over time, we used a Nikon A1$R$ + inverted laser-scanning confocal microscope maintained at 30 ± 1°C. In each experiment, we acquired vertical stacks of planar fluorescence images separated by 2.58 μm along the vertical ($z$) direction, successively every 2–30 min for up to 20 hr. We then produced a maximum intensity projection from each stack at every time frame with the logarithm of fluorescent intensities displayed at every pixel; examples are shown in *Figure 1*. Our prior work used high-resolution visualization to obtain magnified views of the bacterial concentration fields at long times for unperturbed flat fronts and verified that the cells are swimming through the pore space as a suspension (***Bhattacharjee et al., 2021***). Here, we instead use lower-resolution visualization to characterize population-scale front dynamics over larger length scales. We note that because our experiments probe fluorescence from GFP-expressing cells, the confocal images only show the actively moving cells near the leading edge of each propagating front because it is exposed to sufficient oxygen for the GFP to properly fold. As these cells move outward, they continually consume nutrient and oxygen—eventually causing the trailing 'inner' region of the population to become oxygen-depleted, as shown in our previous work (***Bhattacharjee et al., 2021***). Under these conditions, we conjecture that the GFP expressed by the cells does not properly fold, and the cells lose fluorescence over ~30 min. Thus, even though some cells remain localized within the inner region, they turn dark and hence seem to disappear from the microscope fluorescence images.

## Characterizing experimental front dynamics

We used each maximum intensity projection at each time point to manually measure the time-dependent amplitude ($A$) and radial location of the front ($R_f$) as defined in *Figure 1B*, identifying the edges of the front as the positions at which the fluorescent signal from cells matches the background noise.

As we showed in our previous work (***Bhattacharjee et al., 2021***), due to the initially high cell density in the population, inter-cell collisions limit outward migration of the population; a coherent outward-propagating front only forms after at least ~1 hr. Here, we do not focus on these initial transient dynamics, but instead examine the long-time smoothing behavior of undulated fronts. We did this by tracking the decay of the time-dependent undulation amplitude over time, as shown in *Figure 1F*; we identified the time $t_0$ at which smoothing is initiated as the earliest time at which the error associated with an exponential fit to the decay of $A(t)$ is minimized. The initial value $A_0$ is then given by $A(t_0)$.

## Details of continuum model

To mathematically model the dynamics of bacterial fronts, we use a continuum description of chemotactic migration that we previously showed captures the essential dynamical features of flat fronts (***Bhattacharjee et al., 2021***). This model extends previous work on the classic Keller–Segel model (***Fu et al., 2018***; ***Saragosti et al., 2011***; ***Cremer et al., 2019***; ***Croze et al., 2011***; ***Keller and Odell, 1975***; ***Keller and Segel, 1971***; ***Keller and Segel, 1970***; ***Odell and Keller, 1976***; ***Lauffenburger, 1991***; ***Seyrich et al., 2019***) to the case of dense populations in porous media. In particular, we consider a 2D representation of the population in the $xy$ plane for simplicity and describe the evolution of the nutrient concentration $c(\vec{r}, t)$ and number density of bacteria $b(\vec{r}, t)$ using the coupled Equations 1 and 2.

### Nutrient diffusion and consumption

The media used in our experiments have $L$-serine as the most abundant nutrient source and chemoattractant (***Neidhardt et al., 1974***). *E. coli* consume this amino acid first (***Yang et al., 2015***) and respond to it most strongly as a chemoattractant compared to other components of the media (***Wong et al., 2014***; ***Mesibov and Adler, 1972***; ***Adler, 1966***; ***Menolascina et al., 2017***). Furthermore, the nutrient levels of our liquid medium are nearly two orders of magnitude larger than the levels under which *E. coli* excrete appreciable amounts of their own chemoattractant (***Budrene and Berg, 1991***)

and generate strikingly different front behavior (**Budrene and Berg, 1991**; **Budrene and Berg, 1995**; **Mittal et al., 2003**) than those that arise in our experiments; however, the nutrient levels we use are sufficiently low to avoid toxicity associated with extremely large levels of $L$-serine (**Neumann et al., 2014**). Thus, given all of these reasons, we focus on $L$-serine as the primary nutrient source and attractant, described by the scalar field $c(\vec{r}, t)$. **Equation 1** then relates changes in $c$ to nutrient diffusion and consumption by the bacteria. The nutrient diffusion coefficient $D_c = 800$ μm²/s is given by previous measurements in bulk liquid; we treat nutrient diffusion as being unhindered by the highly swollen hydrogel matrix due to its large internal mesh size. The maximal consumption rate per cell $\kappa = 1.6 \times 10^{-11}$ mM (cell/mL)$^{-1}$ s$^{-1}$ is chosen based on previous measurements (**Croze et al., 2011**), and $g(c) = c/\left(c + c_{1/2}\right)$ describes the influence of nutrient availability relative to the characteristic concentration $c_{1/2} = 1$ μM through Michaelis–Menten kinetics, as established previously (**Cremer et al., 2019**; **Croze et al., 2011**; **Monod, 1949**; **Woodward et al., 1995**; **Shehata and Marr, 1971**). These values yield simulated fronts that we have previously validated against experiments in porous media for the unperturbed case (**Bhattacharjee et al., 2021**).

## Bacterial diffusion and chemotaxis

The bacterial flux $\vec{J}_b$ as included in **Equation 2** arises from the undirected and directed motion of cells, that is, diffusion $-D_b \nabla b$ and chemotaxis $b\chi\nabla f(c)$, respectively. Our analysis focuses on the contribution of flagella-mediated swimming through the pore space to these fluxes, following our previous work (**Bhattacharjee et al., 2021**); however, incorporating other possible modes of motility such as surface-assisted propulsion would be a useful future extension. It is generally challenging to calculate the motility coefficients $D_b$ and $\chi$ a priori for a given porous medium and pore size distribution. To circumvent this uncertainty and avoid ad hoc approximations, our simulations use values of both that were directly determined from our prior experiments (**Bhattacharjee and Datta, 2019a**; **Bhattacharjee et al., 2021**).

In particular, we measured the value of the active cellular diffusion coefficient $D_b$ by quantifying the undirected spreading of a dilute population of the same cells in porous media identical to those used here (**Bhattacharjee and Datta, 2019a**), but under uniform nutrient conditions. The measured $D_b = 2.32$, 0.93, and 0.42 μm²/s for porous media with $\xi = 2.2$, 1.7, and 1.2 μm, respectively.

For directed spreading, we describe cellular chemotaxis using the sensing function $f(c) \equiv \log\left[\left(1 + c/c_-\right)/\left(1 + c/c_+\right)\right]$ and the chemotactic coefficient $\chi$, as established previously (**Fu et al., 2018**; **Cremer et al., 2019**). The characteristic concentrations $c_- = 1$ μM and $c_+ = 30$ μM represent the dissociation constants of the nutrient for the inactive and active conformations of the cell-surface receptors, respectively (**Cremer et al., 2019**; **Fu et al., 2018**; **Dufour et al., 2014**; **Yang et al., 2015**; **Sourjik and Wingreen, 2012**; **Shimizu et al., 2010**; **Tu et al., 2008**; **Kalinin et al., 2009**; **Shoval et al., 2010**; **Lazova et al., 2011**; **Celani et al., 2011**). To measure the active chemotactic coefficient $\chi$, we performed the same experiments as reported here, but using flat, unperturbed inocula (**Bhattacharjee et al., 2021**). We then tracked the leading-edge position of the resulting flat chemotactic fronts over time to directly measure the long-time front propagation speed. Then, we performed the same simulations as reported in this paper, but using the flat, unperturbed inocula as initial conditions, and measured the long-time front propagation speed in simulations. Finally, we repeated the simulations for different input values of $\chi$ and used the value of $\chi$ that yielded the best fit with the experimentally measured long-time front propagation speed. Thus, our simulations employ values of $\chi$ that are directly matched to experiments in identically prepared porous media. Similar to $D_b$, the value of $\chi$ decreases with increasing pore-scale confinement (**Bhattacharjee et al., 2021**); we obtained $\chi = 145$, 9, and 5 μm²/s for porous media with $\xi = 2.2$, 1.7, and 1.2 μm, respectively. Although heterogeneity in $D_b$ and $\chi$ may be present within each population itself (**Fu et al., 2018**; **Bai et al., 2021**), we focus our analysis on the influence of pore size by assuming a constant value of both for each simulation.

Finally, we note that the motility parameters $D_b$ and $\chi$ reflect the ability of cells to move through the pore space via an unbiased or biased random walk with mean step length $l$ whose value depends on pore-scale confinement and possible cell-cell collisions in the pore space. For the case of sufficiently dilute cells in porous media, $l$ is set by the geometry of the pore space, as we previously established (**Bhattacharjee and Datta, 2019a**; **Bhattacharjee and Datta, 2019b**); in particular, $l \approx l_c$, the mean length of chords, or straight paths that fit in the pore space (**Torquato and Lu, 1993**). However, when the cells are sufficiently dense, as arises in the experiments explored here, cell-cell collisions truncate

*l*. Indeed, our porous media are highly confining; the pore sizes are <8 µm, comparable to the size of a single-cell body (length ~2 µm and width ~0.5 µm) and its flagella (length ~5 µm), in all cases. Hence, because the pore space is too small to fit multiple cells side-by-side, cell-cell interactions are necessarily restricted to end-on interactions. We directly confirmed this phenomenon experimentally in our prior study (*Bhattacharjee et al., 2021*). This feature of confinement in a tight porous medium is starkly different from the case of cells in homogeneous liquid, in which short-range side-by-side interactions promote alignment of cell clusters and result in cooperative motions at high cell densities. We therefore did not incorporate such cooperative interactions in our model. Instead, we model cell-cell collisions by considering the mean separation between cells $l_{cell} \approx \left( \frac{3f}{4\pi b} \right)^{1/3} - d$, where $f$ is the volume fraction of the pore space between hydrogel particles, $b$ is the local bacterial number density, and $d \approx 1$ µm is the characteristic size of a cell; for simplicity, when $l_{cell} < l_c$, we assume that cell-cell collisions truncate the mean step length $l$ and set its value to $l_{cell}$. That is, wherever $b$ is so large that $l_{cell} < l_c$, we multiply the values of both $D_b$ and $\chi$ used in *Equation 2* by the correction factor $(l_{cell}/l_c)^2$ that accounts for the truncated $l$ due to cell-cell collisions. Moreover, wherever $b$ is even so large that this correction factor is less than zero—that is, cells are jammed—we set both $D_b$ and $\chi$ to zero. Based on our experimental characterization of pore space structure (*Bhattacharjee and Datta, 2019b*), we use $f$ = 0.36, 0.17, and 0.04, and $l_c$ = 4.6, 3.1, and 2.4 µm, for porous media with $\xi$ = 2.2, 1.7, and 1.2 µm, respectively. In this simple mean-field treatment of cell-cell interactions, as the cellular density increases, and thus the mean spacing between cells decreases, they increasingly truncate each other's motion and the motility parameters $D_b$ and $\chi$ decrease—eventually becoming zero when the cells are so densely packed that they do not have space to move.

## Bacterial proliferation

Changes in $b$ can also arise from net cell proliferation, as described in *Equation 2*. In particular, we describe net cell proliferation with the maximal rate per cell $\gamma$ multiplied by the Michaelis–Menten function $g(c)$ that again describes the influence of nutrient availability, that is, it quantifies the reduction in proliferation rate when nutrient is sparse. We directly measured $\gamma \equiv \ln 2/\tau_2$ previously, where $\tau_2 = 60$ min is the mean cell division time in a porous medium for our experimental conditions. We note that because $c$ and $b$ are coupled in our model, we do not require an additional 'carrying capacity' of the population to be included, as is often done (*Cremer et al., 2019*; *Croze et al., 2011*); we track nutrient deprivation directly through the radially symmetric nutrient field $c(\vec{r}, t)$.

## Implementation of numerical simulations

While the experimental geometry is three dimensional, in previous work (*Bhattacharjee et al., 2021*), we found that radial and out-of-plane effects do not need to be considered to capture the essential features of bacterial front formation and migration. Thus, for simplicity, we use a 2D representation. In the $x$ direction (coordinates defined in *Figures 2 and 4*), no flux boundary conditions are used at the walls of the simulated region for both field variables $b$ and $c$. In the $y$ direction, no flux boundary conditions are used after one wavelength of the undulation, peak to peak, which comprises a single repeatable unit. The initial cylindrical distribution of cells 3D printed in the experiments has a diameter of ~100 µm; so, in the $x$ dimension of the numerical simulations, we use a Gaussian with a 100 µm full width at half maximum for the initial bacteria distribution $b(x, t = 0)$, with a peak value that matches the 3D-printed cell density in the experiments, $0.95 \times 10^{12}$ cells/mL. We vary the center $x$ position of the Gaussian distribution sinusoidally along $y$ to reproduce a given experimental wavelength and amplitude. Experimental wavelengths were measured directly from confocal images and rounded to the nearest 10 µm. The initial condition of nutrient is $c = 10$ mM everywhere, characteristic of the liquid media used in the experiments. The initial nutrient concentration is likely lower within the experimental population initially due to nutrient consumption during the 3D printing process; however, we expect this discrepancy to play a negligible role as nutrient deprivation occurs rapidly in the simulations.

As previously detailed (*Bhattacharjee et al., 2021*), while the periphery of a 3D-printed bacterial population forms a propagating front, cells in the inner region remain fixed and eventually lose fluorescence because they are oxygen-limited. Specifically, the fluorescence intensity of this fixed inner population remains constant over an initial duration $\tau_{delay} = 2$ hr, and then exponentially decreases with a decay time scale $\tau_{starve} = 29.7$ min. To facilitate comparison to the experiments, our simulations

incorporate this feature to represent the cellular signal, which is the analog of the fluorescence measured in experiments, in *Figures 2 and 4*. We do this by multiplying the cellular density obtained by solving *Equation 2* by a correction factor that incorporates the history of oxygen depletion. Specifically, wherever $c(\vec{r}\,', t')$ drops below a threshold value, for all times $t > t' + \tau_{\text{delay}}$, we multiply the cellular density $b(\vec{r}\,', t)$ by $e^{-(t-t')/\tau_{\text{starve}}}$, where $t'$ is the time at which the position $\vec{r}\,'$ became nutrient-depleted; oxygen and nutrient depletion occur at similar positions and times as detailed in *Bhattacharjee et al., 2021*.

To numerically solve the continuum model, we use an Adams–Bashforth–Moulton predictor corrector method (*Seyrich et al., 2019*), where the order of the predictor and corrector are 3 and 2, respectively. Since the predictor corrector method requires past time points to inform future

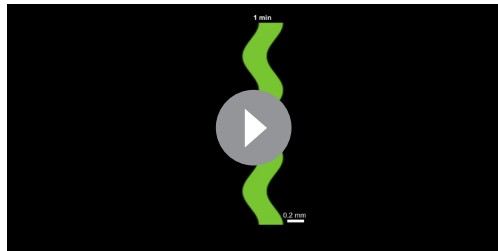

**Video 7.** Simulation probing the condition of lower initial overall nutrient concentration for $\lambda = 0.8$ mm, $\xi = 1.7$ μm. Video shows the calculated cellular fluorescence signal of cells failing to migrate as a front from an undulated stripe of closely packed *E. coli* with initial $c = 10$ μM. To more clearly show the lack of front formation, in this visualization we neglect fluorescence signal loss that occurs when cells are oxygen depleted.
https://elifesciences.org/articles/71226/figures#video7

steps, the starting time points must be found with another method; we choose the Shanks starter of order 6 (*Shanks, 1966*). For the first and second derivatives in space, we use finite difference equations with central difference forms in 2D. Time steps of the simulations are 0.01 s and spatial resolution is 10 μm. Because the experimental chambers are 3.5 cm in diameter, we use a distance of $3.5 \times 10^4$ μm for the size of the entire simulated system in the $x$ direction with the cells initially situated in the center. Our previous work (*Bhattacharjee et al., 2021*) demonstrated that the choice of discretization does not appreciably influence the results in numerical simulations of flat fronts; furthermore, our new results for the simulations performed here (*Figure 4—figure supplement 3*) indicate that our choice of discretization used is sufficiently finely resolved such that the results in numerical simulations of undulated fronts are not appreciably influenced by discretization.

## Characterizing simulated front dynamics

For the analysis shown in *Figure 2*, the leading edge is defined as the locus of positions at which $b$ falls below a threshold value equal to $10^{-4}$ times the maximum cell density of the initial bacterial distribution, as in *Bhattacharjee et al., 2021*. For the analysis shown in *Figure 4*, to more accurately track the leading edge of the front, we define it as the locus of positions at which $b$ falls below a threshold value specific to each condition tested; the threshold is 0.003 cells per μm³ for the prototypical case of $\xi = 1.7$ μm and $\lambda = 0.8$ mm shown in *Figure 4A–C*, as well as all simulations for $\xi = 2.2$ μm; 0.002 cells per μm³ for simulations for $\xi = 1.7$ μm and $\lambda = 2.0$ and 3.2 mm; and 0.001 cells per μm³ for simulations for $\xi = 1.2$ μm and $\lambda = 0.8$ mm. We note that the $b$-dependence of the motility parameters $D_b$ and $\chi$ does not play an appreciable role in our analysis of smoothing since the definition used for the leading edge of each front is at a fixed, low value of $b$.

## Robustness of front smoothing

One may speculate that smoothing could be avoided or even reversed by lowering the initial nutrient concentration to a value in between $c_+$ and $c_-$, thereby diminishing the difference in chemotactic response between peaks and valleys and allowing the amplifying effects of the nutrient gradient to dominate. However, a simulation performed with a much lower initial nutrient concentration of 10 μM throughout, chosen to be in between $c_+$ and $c_-$, does not even form a traveling front at all over the experimental time scale (*Video 7*). This absence of a front is due to the reduction in nutrient consumption as modulated by the Monod function $g(c)$, which results in a drastic reduction in the nutrient gradient that drives front formation and propagation. Thus, despite varying the initial nutrient concentration over three orders of magnitude, the upper limit $c_+$ over an order of magnitude, and the migration parameters $D_b$ and $\chi$ over an order of magnitude, we have not found conditions under which chemotactic fronts, if they form, do not smooth. Smoothing therefore appears to be robust to large changes in the environmental conditions.

## Acknowledgements

It is a pleasure to acknowledge Tommy Angelini for providing microgel polymers; Bob Austin for providing fluorescent *E. coli*; and Stas Shvartsman, Howard Stone, Sankaran Sundaresan, and Ned Wingreen for stimulating discussions. This work was supported by NSF grant CBET-1941716, the Project X Innovation fund, a distinguished postdoctoral fellowship from the Andlinger Center for Energy and the Environment at Princeton University to TB, the Eric and Wendy Schmidt Transformative Technology Fund at Princeton, the Princeton Catalysis Initiative, the Pew Charitable Trusts through the Pew Biomedical Scholars Program, and in part by funding from the Princeton Center for Complex Materials, a Materials Research Science and Engineering Center supported by NSF grant DMR-2011750. This material is also based upon work supported by the National Science Foundation Graduate Research Fellowship Program (to J.A.O.) under Grant No. DGE-1656466. Any opinions, findings, and conclusions or recommendations expressed in this material are those of the authors and do not necessarily reflect the views of the National Science Foundation. RA acknowledges support from the Human Frontier Science Program (LT000475/2018C).

## Additional information

### Competing interests

Tapomoy Bhattacharjee, Sujit Sankar Datta: The experimental platform used to 3D print and image bacterial communities in this publication is the subject of a patent application filed by Princeton University on behalf of T.B. and S.S.D. (PCT Application number PCT/US/2020/030213). The other authors declare that no competing interests exist.

### Funding

| Funder | Grant reference number | Author |
| --- | --- | --- |
| National Science Foundation | CBET-1941716 | Sujit Sankar Datta |
| National Science Foundation | DMR- 2011750 | Sujit Sankar Datta |
| National Science Foundation | DGE-1656466 | Jenna Anne Ott |
| Project X Innovation fund | | Sujit Sankar Datta |
| Andlinger Center for Energy and the Environment | Distinguished Postdoctoral Fellowship | Tapomoy Bhattacharjee |
| Eric and Wendy Schmidt Transformative Technology Fund | | Sujit Sankar Datta |
| Human Frontier Science Program | LT000475/2018-C | Ricard Alert |
| Princeton Catalysis Initiative | | Sujit Sankar Datta |
| Pew Charitable Trusts | Pew Biomedical Scholars Program | Sujit Sankar Datta |

The funders had no role in study design, data collection and interpretation, or the decision to submit the work for publication.

### Author contributions

Tapomoy Bhattacharjee, Data curation, Formal analysis, Investigation, Validation, Visualization, Writing – original draft; Daniel B Amchin, Data curation, Formal analysis, Investigation, Methodology, Software, Validation, Visualization, Writing – original draft; Ricard Alert, Formal analysis, Methodology, Writing – original draft, Writing – review and editing; Jenna Anne Ott, Investigation, Methodology,

Software, Writing – original draft; Sujit Sankar Datta, Conceptualization, Funding acquisition, Methodology, Project administration, Resources, Supervision, Visualization, Writing – original draft, Writing – review and editing

**Author ORCIDs**
Tapomoy Bhattacharjee (iD) http://orcid.org/0000-0001-8899-1379
Daniel B Amchin (iD) http://orcid.org/0000-0002-1557-0984
Ricard Alert (iD) http://orcid.org/0000-0002-1885-9177
Jenna Anne Ott (iD) http://orcid.org/0000-0001-6832-0658
Sujit Sankar Datta (iD) http://orcid.org/0000-0003-2400-1561

**Decision letter and Author response**
Decision letter https://doi.org/10.7554/eLife.71226.sa1
Author response https://doi.org/10.7554/eLife.71226.sa2

## Additional files

**Supplementary files**
• Transparent reporting form

**Data availability**
All data needed to evaluate the conclusions in the paper are present in the paper and/or the Supplementary Materials, and the raw data have been deposited in the publicly- accessible DataSpace repository at https://doi.org/10.34770/3g41-6j28.

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
