## [Editor Report]

The work provides new insights into the dual role of chemotactic sensing in both generating and controlling bacterial wave front patterns. Novel and elegant experimental techniques supported by computations using phenomenological models validate the hypothesis that chemotactic sensing smooths morphological variations; however, experiments suggest a richer picture than that predicted by the theory.

---

## [Decision Letter]

**Decision letter after peer review:**

Thank you for submitting your article "Chemotactic smoothing of collective migration" for consideration by *eLife*. Your article has been reviewed by 2 peer reviewers, and the evaluation has been overseen by a Reviewing Editor and Aleksandra Walczak as the Senior Editor. The reviewers have opted to remain anonymous.

Essential revisions:

Both referees reviewed positively the paper. They however suggest a number of clarifications and a more extended discussion of a few issues. The authors are invited to carefully read the attached full reports and to address all the referees' comments in a revised version.

Particular attention must be devoted to:

– Clarify how certain parameters are estimated from the data (e.g. diffusion coefficient, pore properties); discuss the dependence of the susceptibility on pore size and the potential role of bacterial concentration (referee 2).

– Discuss connections with some growth models potentially relevant for the experimental findings presented in the paper (referee 1).

*Reviewer #1:*

The paper is very clearly written an presents a nice mix of experimental and numerical results. The authors might want to make a connection with the growth-expansion model by Cremer and Honda (see also Narla Cremer and Hwa, arxiv 2103.08100) which appears to be the limiting case of c_{1/2}\to 0 c_{-}\to 0 and infinite carrying capacity. For that model, planar traveling wave solutions exist and their speed and width of the band are analytically known (approximately exact).

It would be interesting to study analytically the transverse stability of such solutions, even though this is clearly outside the scope of the present paper.

*Reviewer #2:*

Collective bacterial motility relies intimately on the manner by which individual cells sense their environment, chemical and physically sense their neighbors, and adjust to appropriate chemical fields such as nutrient concentrations. In favorable conditions, bacterial cells grow, multiply and move together in well defined fronts or waves. There is a large body of literature on how intrinsic (phenotypic) and extrinsic (environmental) perturbations disrupt these patterns and quench collectivity or destabilize propagating fronts. Here, the authors address the complementary question of how propagating bacterial populations may withstand perturbations. A elegant and novel 3D printing platform that allows for the generation localized, dense bacterial populations with controllable mesoscale structure (wavelength and shape) within a hindering hydrogel medium is used to study the long time spatiotemporal dynamics of bacterial collective motility. Experiments show that initial structured interfaces smooth into rapidly moving bacterial fronts that propagate stably. The authors hypothesize that this autonomous and emergent quenching of destabilizing density perturbations is due to chemotactic response with both single cell and collective aspects playing critical roles. A variation of the Keller-Segel model incorporating bacterial growth, nutrient consumption, and bacterial spread due to diffusion and chemotaxis provides predictions that compare well to experiments and support the hypothesis.

Collective motility with diffusive and (concentration-induced) fluxes feature in many active matter systems; thus insights from this work are applicable to sustained pattern formation in these related systems.

Strengths:

A significant strength of the paper is in the synthesis of novel experiments with variations of a well-established theoretical framework to investigate the role of chemotaxis in modulating bacterial front undulations. The hypothesis is clearly articulated with all features of the growing bacterial population – viz, growth, motility and chemotactic sensing considered in understanding the experiments.

The experimental techniques and protocols that allow for direct imaging of bacterial waves and propagating fronts are novel and allow for cleanly printed sinuous patterns. The hydrogel medium allows for both control of the medium porosity (mean pore size), and also for clean control of nutrient concentration. The data obtained is of high quality and clearly exhibits the smoothing of wave fronts that is the focus of the paper. The permeability of the gel particles allows for experiments where local oxygen and nutrient fields may be sensed easily since small scale gradients are prevented.

It is hard to use the experiments for analyzing the roles played by each of these in isolation.Thus the role of analysis is important – specifically, any phenomenological model used should capture the important biophysical factors that may play a role. The continuum model for chemotactic spread of growing bacterial suspensions (via a Keller-Segel type formulation) is well chosen and carefully analyzed with attempts first to negate the hypothesis (by asking if growth, non-chemotactic diffusion based spreading may mimic the dynamics) and then by proactively supporting it. Computations seem to directly include parameters evaluated from experiments and qualitatively capture the nature of the rapidly propagating wavefronts far from the initial positions. Results with diffusion, and cell proliferation knocked out by choosing parameters appropriately suggest that chemotactic response drives front formation, and is implicated in sustained and unperturbed wave propagation. A strength of this manner of interrogation is the ease by which more complicated features may be introduced in the theory and in the computations. This generality in principle allows extensions to more complicated systems that the authors suggest may benefit from the insights from this paper.

The principle that variations in the driving force (gradient of the nutrient concentration, c here) may compete with variations in cell response to the value of the quantity (the value of c) and overall determine the evolution of undulations at the front is interesting and a crucial insight. Active matter systems such as bacterial swarms, multicellular clusters etc exhibit diffusive, nutrient sensing chemotactic behavior and haptotaxis. Insights from this paper may help understand how emergent patterns remain stable and long-lasting in such systems. The results of this paper and insights will contribute to the on-going dialogue as to the complementary role of collective vs cell-level behavior in controlling motility.

The conclusions of this paper are mostly well supported by the experimental augmented by the computational analysis and associated discussions. Some aspects of the presentation may benefit from further clarification.

1) Some discussion is warranted on the role of additional motility modes such as surface assistant motility and the general tendency of bacteria to stick around near surfaces in mediating the non-chemotactic behavior; and possible surface assistant motility that may help bacteria escape in the initial stages of the wave front formation and propagation and prevent residual trapping. Experiments show a rather striking phenomena for some combinations of the wavelengths and pore size – bacteria located at the point of printing disperse and eventually the central region seems to have local density of bacteria with most of the cells participating in the front and moving away rapidly. The model however seems to suggest that a fraction of the initial cells remain localized at their initial position. Some discussion of how the model may be adapted or modified to account for this will help.

2) Some more clarity and discussion is need to explain how experiments are used to calculate the values of parameters in the simulations. The susceptibility function that connects the convective chemotactic induced flux is dependent on the pore-size and hindrance effects. The effect of confinement for a diffusing nutrient molecule is easy to rationalize based on known literature. The bacterial diffusion coefficient may also be anticipated (for a swimming cell) by using concepts from kinetic theory where an effective diffusion coefficient came be estimated by calculating the mean speed between either tumbling events, wall interactions and reorientations, or cell-cell interactions and reorientations. The manner in which Chi may vary with pore size is however unclear and needs some elaboration. Here, the form of the pore size distribution in addition to the mean pore size may be important.

3) Bacterial cell properties need to be mentioned and connected to pore geometry. Specifically, the cells are stated to have a characteristic size. But bacteria are rods with high aspect rations. Given this, does the concentration of the cells correspond to a semi-dilute or concentrated suspension. If sufficiently concentration alignment effects may arise that augment and enhance over-all spread in addition to chemotaxis based flux.

I encourage the authors to address the following comments in their response or revision.

1) The pore size is a crucial feature in the experiments. However only the mean pore size is reported. It would help to have an estimate of the pore size distribution.

2) Close ups of the bacterial concentration fields in Figure 1(C,D,E) would be very helpful. While the t=0 case is probably difficult to show due to the rather high density of bacteria, how about the large t values ? The experiments do suggest that the bacteria are moving in suspension and not along the gel surfaces. Is this something that can be checked by looking at the suspensions at higher resolutions?

3) The analysis of chemotaxis (following equation 3, and subsequent discussion) brings up an important fact. The physics involves two competing effects – the spatial variation in c that actually amplifies front undulations and irregularities and the intrinsic chemotactic response that allows for receptors to saturate. The form of the response function f(c) and the form of Chi used in the paper is relevant to *E. coli*. Can the authors further comment on if and how one may modify f(c) and chi to suppress or enhance these effects. There is discussion of this in the final section – however a sample computation that illustrates this would emphasize this point.

4) There are some experimental details missing in the paper. How was the diffusion coefficient D_b measured? For a dense suspension of bacteria (without chemotactic effects), the effective long-time bacterial diffusivity may depend on the concentration of bacteria due to cell-cell interactions and alignment in a manner seen in dense suspensions of rods. Was this done in the dilute regime by tracking single bacteria MSD or was this done when the concentration of bacteria was high?

5) The discussion on calculating the mean separation between cells in the methods section is a tad confusing and needs clarity. A recapitulation or summary of the results from the previous publication would help here.

---

## [Author Response]

Essential revisions:Both referees reviewed positively the paper. They however suggest a number of clarifications and a more extended discussion of a few issues. The authors are invited to carefully read the attached full reports and to address all the referees' comments in a revised version.Particular attention must be devoted to:– Clarify how certain parameters are estimated from the data (e.g. diffusion coefficient, pore properties); discuss the dependence of the susceptibility on pore size and the potential role of bacterial concentration (referee 2).– Discuss connections with some growth models potentially relevant for the experimental findings presented in the paper (referee 1).Reviewer #1:The paper is very clearly written an presents a nice mix of experimental and numerical results. The authors might want to make a connection with the growth-expansion model by Cremer and Honda (see also Narla Cremer and Hwa, arxiv 2103.08100) which appears to be the limiting case of c_{1/2}\to 0 c_{-}\to 0 and infinite carrying capacity. For that model, planar traveling wave solutions exist and their speed and width of the band are analytically known (approximately exact).It would be interesting to study analytically the transverse stability of such solutions, even though this is clearly outside the scope of the present paper.

We are grateful to the Reviewer for their positive assessment of our work, the time they took to read through our work, and their constructive feedback. Indeed, the growth-expansion model of Cremer, Honda, *et al.* and the related work by Narla, Cremer, and Hwa represents an important recent development in the study of chemotaxis, and we have now cited both in the revised manuscript and discussed potential connections with our own work. As the Reviewer insightfully noted, the growth-expansion model can be thought of as a limit of our Keller-Segel model, in which bacterial growth is taken to be independent of the attractant.

We fully agree with the Reviewer that, while it is outside the scope of the present paper, it would be interesting to study analytically the transverse stability of the traveling wave solutions of the growth-expansion model. We have now explicitly noted this point in the revised manuscript. Furthermore, we note that in more recent work (preprint available at arxiv 2107.11702), we have studied analytically the transverse linear stability of chemotactic fronts using a simplified version of our Keller-Segel model. Extending this analysis to the case of the growth-expansion model would be a very interesting direction for future work. Motivated by the Reviewer’s comment, we are now exploring this extended analysis in ongoing work.

Reviewer #2:Collective bacterial motility relies intimately on the manner by which individual cells sense their environment, chemical and physically sense their neighbors, and adjust to appropriate chemical fields such as nutrient concentrations. In favorable conditions, bacterial cells grow, multiply and move together in well defined fronts or waves. There is a large body of literature on how intrinsic (phenotypic) and extrinsic (environmental) perturbations disrupt these patterns and quench collectivity or destabilize propagating fronts. Here, the authors address the complementary question of how propagating bacterial populations may withstand perturbations. A elegant and novel 3D printing platform that allows for the generation localized, dense bacterial populations with controllable mesoscale structure (wavelength and shape) within a hindering hydrogel medium is used to study the long time spatiotemporal dynamics of bacterial collective motility. Experiments show that initial structured interfaces smooth into rapidly moving bacterial fronts that propagate stably. The authors hypothesize that this autonomous and emergent quenching of destabilizing density perturbations is due to chemotactic response with both single cell and collective aspects playing critical roles. A variation of the Keller-Segel model incorporating bacterial growth, nutrient consumption, and bacterial spread due to diffusion and chemotaxis provides predictions that compare well to experiments and support the hypothesis.Collective motility with diffusive and (concentration-induced) fluxes feature in many active matter systems; thus insights from this work are applicable to sustained pattern formation in these related systems.Strengths:A significant strength of the paper is in the synthesis of novel experiments with variations of a well-established theoretical framework to investigate the role of chemotaxis in modulating bacterial front undulations. The hypothesis is clearly articulated with all features of the growing bacterial population – viz, growth, motility and chemotactic sensing considered in understanding the experiments.The experimental techniques and protocols that allow for direct imaging of bacterial waves and propagating fronts are novel and allow for cleanly printed sinuous patterns. The hydrogel medium allows for both control of the medium porosity (mean pore size), and also for clean control of nutrient concentration. The data obtained is of high quality and clearly exhibits the smoothing of wave fronts that is the focus of the paper. The permeability of the gel particles allows for experiments where local oxygen and nutrient fields may be sensed easily since small scale gradients are prevented.It is hard to use the experiments for analyzing the roles played by each of these in isolation. Thus the role of analysis is important – specifically, any phenomenological model used should capture the important biophysical factors that may play a role. The continuum model for chemotactic spread of growing bacterial suspensions (via a Keller-Segel type formulation) is well chosen and carefully analyzed with attempts first to negate the hypothesis (by asking if growth, non-chemotactic diffusion based spreading may mimic the dynamics) and then by proactively supporting it. Computations seem to directly include parameters evaluated from experiments and qualitatively capture the nature of the rapidly propagating wavefronts far from the initial positions. Results with diffusion, and cell proliferation knocked out by choosing parameters appropriately suggest that chemotactic response drives front formation, and is implicated in sustained and unperturbed wave propagation. A strength of this manner of interrogation is the ease by which more complicated features may be introduced in the theory and in the computations. This generality in principle allows extensions to more complicated systems that the authors suggest may benefit from the insights from this paper.The principle that variations in the driving force (gradient of the nutrient concentration, c here) may compete with variations in cell response to the value of the quantity (the value of c) and overall determine the evolution of undulations at the front is interesting and a crucial insight. Active matter systems such as bacterial swarms, multicellular clusters etc exhibit diffusive, nutrient sensing chemotactic behavior and haptotaxis. Insights from this paper may help understand how emergent patterns remain stable and long-lasting in such systems. The results of this paper and insights will contribute to the on-going dialogue as to the complementary role of collective vs cell-level behavior in controlling motility.The conclusions of this paper are mostly well supported by the experimental augmented by the computational analysis and associated discussions. Some aspects of the presentation may benefit from further clarification.

We are grateful to the Reviewer for their positive assessment of our work, the time they took to read through our work, and their constructive feedback. Their summary of our work is beautifully written and nicely highlights the novelty and importance of our work within the broader context of studies of active matter and cellular motility. We are particularly encouraged that the Reviewer highlighted that our experiments are “novel and elegant” and the theory and simulations are “well chosen and carefully analyzed”.

Furthermore, we appreciate the Reviewer’s insightful comments—all of which we fully agree with, and all of which we have now addressed in the revised manuscript (detailed below), guided by the Reviewer’s helpful suggestions. Indeed, several aspects of our discussion were investigated in detail in our prior study [published in *Biophysical Journal,* 120, 3483 (2021) and cited in the present manuscript as Ref. 32], which utilized porous media and cells that were prepared identically to those used in the present paper, but for the case of unperturbed flat chemotactic fronts. We therefore did not include these details in the present manuscript. However, to directly address the Reviewer’s questions, we have now provided more details of these points in the revised manuscript, and thank the Reviewer for guiding us to do so. These revisions have greatly improved the clarity of our manuscript.

1) Some discussion is warranted on the role of additional motility modes such as surface assistant motility and the general tendency of bacteria to stick around near surfaces in mediating the non-chemotactic behavior; and possible surface assistant motility that may help bacteria escape in the initial stages of the wave front formation and propagation and prevent residual trapping. Experiments show a rather striking phenomena for some combinations of the wavelengths and pore size – bacteria located at the point of printing disperse and eventually the central region seems to have local density of bacteria with most of the cells participating in the front and moving away rapidly. The model however seems to suggest that a fraction of the initial cells remain localized at their initial position. Some discussion of how the model may be adapted or modified to account for this will help.

The Reviewer raises an important point. We appreciate having the chance to better clarify the dynamics of the bacteria located at the inner central region of the initial inoculum. We explicitly investigated these dynamics in our previous work on flat chemotactic fronts, but did not adequately clarify this behavior in this manuscript and appreciate having the chance to do so.

As the Reviewer astutely noted, the model suggests that a fraction of the cells remain localized at their initial position, while in the experiments the bacteria appear to move away. In fact, even in the experiments, a fraction of the cells located in the inner central region also remain localized, due to the high local cell density—which impedes outward motion through increased cell-cell collisions—and eventual loss of nutrient and oxygen due to consumption by cells at the periphery. We directly verified this phenomenon in our prior study of flat (unperturbed) fronts [cited in the present manuscript as Ref. 32]: see, for example, Figure S2 in that prior paper.

As cells at the periphery move outward, they continually consume nutrient and oxygen—eventually causing the inner region to become oxygen-depleted, as corroborated by our numerical simulations. Under these conditions, the green fluorescent protein (GFP) expressed by the cells does not properly fold, and the cells lose fluorescence over the measured time scale of τ_starve_ = 29.7 min. Thus, even though some cells remain localized within the inner central region, they turn dark and hence appear to disappear from the microscope fluorescence images. Only the outward-moving cells at the periphery of the front remain bright and appear in the images. To account for this phenomenon, the simulation results shown in Figures 2-4 have a correction factor that incorporates the history of oxygen depletion that is directly motivated by the experimental characterization. We have now explicitly clarified these details in the main text of the manuscript, and appreciate the Reviewer’s guidance in helping us to do so. Moreover, as the Reviewer noted, additional motility modes such as surface-assisted motility could also aid cells in escaping this inner central region and prevent residual trapping; we have now discussed this possibility in the revised manuscript as well.

2) Some more clarity and discussion is need to explain how experiments are used to calculate the values of parameters in the simulations. The susceptibility function that connects the convective chemotactic induced flux is dependent on the pore-size and hindrance effects. The effect of confinement for a diffusing nutrient molecule is easy to rationalize based on known literature. The bacterial diffusion coefficient may also be anticipated (for a swimming cell) by using concepts from kinetic theory where an effective diffusion coefficient came be estimated by calculating the mean speed between either tumbling events, wall interactions and reorientations, or cell-cell interactions and reorientations. The manner in which Chi may vary with pore size is however unclear and needs some elaboration. Here, the form of the pore size distribution in addition to the mean pore size may be important.

We appreciate the Reviewer’s thoughtful comment, and fully agree with them that it is unclear how the chemotactic coefficient and varies with pore size (and pore size distribution). Indeed, to circumvent this uncertainty and avoid *ad hoc* approximations, our simulations use values of the bacterial diffusion coefficient D_b_ and chemotactic coefficient that were determined from experiments. The details were presented in our prior study of flat fronts [Ref. 32] and we therefore did not include them in the present manuscript. However, guided by the Reviewer’s comment, we have now provided more details on how the simulation parameters were obtained from experiments in the revised manuscript as well.

In particular, to measure the chemotactic coefficient , we performed the same experiments as reported in the present paper, but using flat, unperturbed inocula. We then tracked the leading-edge position of the resulting flat chemotactic fronts over time to directly measure the long-time front propagation speed. Then, we performed the same simulations as reported in the present paper, but using the flat, unperturbed inocula as initial conditions, and measured the long-time front propagation speed in simulations. Finally, we repeated the simulations for different input values of , and used the value of that yielded the best fit with the experimentally-measured longtime front propagation speed. Thus, our simulations employ values of that are directly matched to experiments in identically-prepared porous media. We have now explicitly clarified these details in the revised manuscript, and appreciate the Reviewer’s encouragement to do so.

3) Bacterial cell properties need to be mentioned and connected to pore geometry. Specifically, the cells are stated to have a characteristic size. But bacteria are rods with high aspect rations. Given this, does the concentration of the cells correspond to a semi-dilute or concentrated suspension. If sufficiently concentration alignment effects may arise that augment and enhance over-all spread in addition to chemotaxis based flux.

The Reviewer raises an interesting point that we appreciate having the chance to clarify. We fully agree with the reviewer that the bacterial cell shape, aspect ratio, and local cell concentration in the pore space needs to be noted. We had included some of these details in our prior study of flat fronts [Ref. 32], and therefore did not include them in the present manuscript. However, guided by the Reviewer’s comment, we have now provided more details on the cellular geometry and local concentration in the revised manuscript.

In particular, we note that our porous media are highly confining; the pore size is always < 8 μm (see comment on pore size distributions below), comparable to the size of a single cell body and its flagella, in all cases. Thus, because the pore space is too small to fit multiple cells side-by-side, cell-cell interactions are necessarily restricted to endon interactions. We directly confirmed this phenomenon experimentally in our prior study [Ref. 32]. For example, we visualized the interaction between two isolated cells, whose bodies are shown in green and have fluorescently labeled flagella (which help determine the orientation of the moving cell) shown in magenta in the sequence of time-lapse images in Author response image 1. One cell (initially in the top right) is moving in the pore space, while the other (middle) is trapped for the duration of the video. The moving cell eventually collides with the initially trapped cell (second frame), causing it to become transiently trapped itself (third frame), until it can eventually reorient and continue to move through the pore space (fourth frame). Thus, this end-on interaction truncates the hopping length of the moving cell.

**Author response image 1. sa2fig1:** 

Other examples of similar collisions between cells, and lack of any other cooperative motions, are provided in Ref. 32. This feature of confinement in a tight porous medium is starkly different from the case of cells in homogeneous liquid, in which short-range side-by-side interactions promote alignment of cell clusters and result in cooperative motions at high cell densities. We therefore did not incorporate such cooperative interactions in our model. However, as the Reviewer noted, in media with larger pores, alignment effects between neighboring cells could indeed augment and enhance the overall cellular spreading. We have now explicitly clarified these points in the revised manuscript, and thank the Reviewer for their helpful question that guided our efforts.

I encourage the authors to address the following comments in their response or revision.1) The pore size is a crucial feature in the experiments. However only the mean pore size is reported. It would help to have an estimate of the pore size distribution.

We appreciate having the opportunity to provide details of the full pore size distribution. In fact, we previously measured these distributions for porous media prepared identically to those used in the present paper, and included the details in our prior study of flat fronts [Ref. 32]; we therefore did not include them in the present manuscript. Examples of the measured pore size distributions are shown in Figure S1 of Ref. 32. In particular, we tracked the thermal diffusion of 200 nm-diameter fluorescent tracer particles in the pore space; the tracer root mean-squared displacement becomes non-diffusive at a pore size length scale α. Measurements of many different tracers yielded the complementary cumulative distribution function 1-CDF(α) for each of three different porous media tested, shown by the three different colors in the plot. In all three cases, the pore size distribution quantified by 1-CDF(α) is exponential, as shown by the grey lines. Following the Reviewer’s suggestion, we have now explicitly clarified this point in the revised manuscript.

2) Close ups of the bacterial concentration fields in Figure 1(C,D,E) would be very helpful. While the t=0 case is probably difficult to show due to the rather high density of bacteria, how about the large t values ? The experiments do suggest that the bacteria are moving in suspension and not along the gel surfaces. Is this something that can be checked by looking at the suspensions at higher resolutions?

We thank the Reviewer for this question. While in this study we did not perform high-resolution visualization to obtain close ups of the bacterial concentration fields at long times, we did so in great detail in our prior study [Ref. 32], which utilized porous media and cells that were prepared identically to those used in the present paper, but for the case of unperturbed flat fronts. We therefore did not include these details in the present manuscript. However, guided by the Reviewer’s question, we have now provided more details of this point in the revised manuscript.

In particular, as the Reviewer insightfully noted, the experiments do suggest that the bacteria are moving in suspension and not along the gel surfaces. Our prior experimental visualization (e.g., Video S9 in Ref. 32) directly demonstrated this point for similar chemotactic fronts. Indeed, a sample trajectory of one cell moving in such a front over a duration of 14.9 s is shown to the right (adapted from Figure 3 of Ref. 32). The scale bar represents 10 μm.

3) The analysis of chemotaxis (following equation 3, and subsequent discussion) brings up an important fact. The physics involves two competing effects – the spatial variation in c that actually amplifies front undulations and irregularities and the intrinsic chemotactic response that allows for receptors to saturate. The form of the response function f(c) and the form of Chi used in the paper is relevant to *E. coli*. Can the authors further comment on if and how one may modify f(c) and chi to suppress or enhance these effects. There is discussion of this in the final section – however a sample computation that illustrates this would emphasize this point.

We thank the Reviewer for bringing up this important point and encouraging us to illustrate it. As we describe in the revised Discussion, the sensing function (c) must be concave, (c)<0, for chemotactic smoothing to take place. For example, this condition is fulfilled if the sensing function (c) saturates as c increases. To illustrate the necessity of this condition as suggested by the reviewer, we considered a strictly linear sensing function f(c)=cclin, which does not saturate. In this case, the chemotactic response is independent of concentration, f′(c)=1clin′ and we have (c)=0. We chose clin= 11/c_−1/c+ so that the linear (c) matches our original logarithmic (c) at small c. We then performed a sample computation to show that the bacterial front does not smoothen with the linear sensing function. Specifically, we computed the chemotactic velocity vc=(c)c along the front, and showed that in this case, valleys no longer move faster than peaks. Instead, the profile of chemotactic velocity is now inverted with respect to that of Figure 4C bottom, as shown in the newly-added Figure S2, and hence the front does not smooth. Overall, this sample computation illustrates a way of modifying (c) that abrogates sensing saturation and hence would prevent chemotactic smoothing.

4) There are some experimental details missing in the paper. How was the diffusion coefficient D_b measured? For a dense suspension of bacteria (without chemotactic effects), the effective long-time bacterial diffusivity may depend on the concentration of bacteria due to cell-cell interactions and alignment in a manner seen in dense suspensions of rods. Was this done in the dilute regime by tracking single bacteria MSD or was this done when the concentration of bacteria was high?(5) The discussion on calculating the mean separation between cells in the methods section is a tad confusing and needs clarity. A recapitulation or summary of the results from the previous publication would help here.

We appreciate both of these comments from the Reviewer, and are responding to them together, as they are closely related. The Reviewer is completely correct that the effective long-time bacterial diffusivity may depend on the bacterial concentration; therefore, our model uses an approximate density-dependent correction to the diffusivity to account for this concentration dependence. In particular, we use tracking of individual cells in the dilute regime to determine the dilute, cell density-independent bacterial diffusivity, as detailed in our prior work published in *Nature Communications,* 10, 2075 (2019), cited as Ref. 30 in the current manuscript. However, as the Reviewer insightfully noted, cell-cell interactions reduce this diffusivity: In particular, our prior experimental visualization of collisions between cells in the pore space indicate that cells truncate each other’s hops through the pore space in a density dependent manner (as detailed in Ref. 32). Thus, as a first step toward incorporating crowding into the classic Keller-Segel model, we adopt a mean-field treatment of cell-cell interactions using a correction factor that truncates the characteristic step lengths of the random walks performed by the cells based on the mean spacing between cells. In this model, as the cellular density increases, and thus the mean spacing between cells decreases, they increasingly truncate each others’ motion and the motility parameters D_b_ and decrease – eventually becoming zero when the cells are so densely packed that they do not have space to move.

Furthermore, we appreciate the Reviewer’s noting that our description of how this mean spacing between cells was calculated was previously unclear. We have now explicitly clarified this point, as well as the details of how the longtime bacterial diffusion coefficient was determined, in the revised manuscript. We are grateful to the Reviewer for their question, which has helped us further clarify this important point.